# Dissecting extracellular and intracellular distribution of nanoparticles and their contribution to therapeutic response by monochromatic ratiometric imaging

Yue Yan[1,2,4], Binlong Chen [1,2,4], Qingqing Yin[2], Zenghui Wang[2], Ye Yang[2], Fangjie Wan[2], Yaoqi Wang[2], Mingmei Tang[2], Heming Xia[2], Meifang Chen[2], Jianxiong Liu[2], Siling Wang[3], Qiang Zhang[1,2] & Yiguang Wang [1,2✉]

Efficient delivery of payload to intracellular targets has been identified as the central principle for nanomedicine development, while the extracellular targets are equally important for cancer treatment. Notably, the contribution of extracellularly distributed nanoparticles to therapeutic outcome is far from being understood. Herein, we develop a pH/light dual-responsive monochromatic ratiometric imaging nanoparticle (MRIN), which functions through sequentially lighting up the intracellular and extracellular fluorescence signals by acidic endocytic pH and near-infrared light. Enabled by MRIN nanotechnology, we accurately quantify the extracellular and intracellular distribution of nanoparticles in several tumor models, which account for 65–80% and 20–35% of total tumor exposure, respectively. Given that the majority of nanoparticles are trapped in extracellular regions, we successfully dissect the contribution of extracellularly distributed nanophotosensitizer to therapeutic efficacy, thereby maximize the treatment outcome. Our study provides key strategies to precisely quantify nanocarrier microdistribtion and engineer multifunctional nanomedicines for efficient theranostics.

[1] State Key Laboratory of Natural and Biomimetic Drugs, Peking University, Beijing 100191, China. [2] Beijing Key Laboratory of Molecular Pharmaceutics and New Drug Delivery Systems, School of Pharmaceutical Sciences, Peking University, Beijing 100191, China. [3] School of Pharmacy, Shenyang Pharmaceutical University, Shenyang, Liaoning 110016, China. [4] These authors contributed equally: Yue Yan, Binlong Chen. ✉email: yiguang.wang@pku.edu.cn

Engineered nanomedicines have received tremendous attention for cancer treatment in the past few decades[1–5]. After intravenous administration, nanomedicines mostly go through the following steps to achieve superior therapeutic efficacy: circulation in the bloodstream, accumulation in the tumor, penetration into deep tumor tissues, and subsequent internalization into tumor cells[6]. Due to the complex tumor microenvironment, only a part of the nanoparticles (NPs) that accumulate at the tumor sites can be endocytosed into tumor cells[7], and the other NPs are all sequestered in the extracellular regions. Given that most known drug targets (e.g., DNA and microtubule) are located in the intracellular compartments, the therapeutic outcomes of nanomedicines greatly depend on cellular internalization efficiency rather than tumor accumulation[8–10]. Therefore, in order to improve therapeutic efficacy of nanoparticles, many efforts have been devoted to increase cellular internalization[11,12]. However, extracellular therapeutic targets (e.g., tenascin C, hyaluronan, fibronectin, collagen and matrix metalloproteinase; cytokines such as M-CSF; cell membrane receptors such as EGFR) are equally important for cancer treatment[13,14]. For example, some studies show that extracellular photodynamic therapy (PDT), such as cell membrane-targeted PDT can lead to membrane disintegration only by a mild treatment[15,16]. Normally, the pharmacokinetics and biodistribution analyses are performed to establish the correlation between the delivery efficiency and therapeutic efficacy of nanomedicines. However, the routine quantitative and imaging methods can only provide their macroscopic distribution in tumor tissues, but fail to provide the detail information on intracellular and extracellular distributions of nanoparticles[17–19].

Currently, several strategies have been developed to quantify the intracellular distribution of nanoparticles by ruling out the contribution of extracellular ones at cellular or ex vivo level, including inductively coupled plasma mass spectrometry (ICP-MS)[7], bioluminescence reaction[20], and chemical etching[21], etc. In addition, microdialysis has been exploited to determine the nanoparticle exposure in extracellular fluid in vivo[22]. Despite the successful quantification of nanoparticle microdistribution in vitro and in vivo, simultaneously quantifying extracellularly and intracellularly distributed nanoparticles, and further parsing their contribution to the therapeutic efficacy in living subjects still remains a big challenge.

Herein, we report a monochromatic ratiometric imaging nanoparticle (MRIN) that can precisely quantify extracellular and intracellular distribution of nanoparticles, enabling the parsing of nanoparticle microdistribution on therapeutic contribution in vivo (Fig. 1). The MRIN is a pH/light dual-responsive nanoparticle fluorescently labeled with a near-infrared fluorophore (e.g., Cy5) and a fluorescence quencher (Cy7.5). MRIN keeps fluorescence signal 'OFF' in the bloodstream and extracellular tumor space (pH$_e$ ~ 6.7–7.1)[23] due to the Förster resonance energy transfer (FRET) effect from Cy5 to Cy7.5 fluorophore. After cellular internalization in tumor tissues, the Cy5 signal of MRIN immediately turns 'ON' due to nanoparticle dissociation in the acidic endo-lysosomal environment (pH$_{ee}$ ~ 6.0)[24,25], whereas the extracellular distributed nanoparticles still keep 'OFF', allowing the quantification of internalized nanoparticles in tumor tissues. Then, an external 808 nm laser irradiation is exploited to photodegrade fluorescence quencher[26], thereby lighting up the Cy5 signals of extracellular distributed nanoparticles in tumor tissues for their accurate quantification. Thus, the intracellularly and extracellularly located nanoparticles in tumor regions are sequentially lighted up by non-crosstalk stimuli for monochromatic ratiometric imaging of nanoparticle microdistribution in living animals. Furthermore, a monochromatic ratiometric imaging/therapeutic nanoparticle (MRITN) is also fabricated to uncover the photodynamic therapeutic (PDT) efficacy of

nanoparticles that resided in extracellular and intracellular regions, and finally maximize the PDT efficacy.

## Results

### Design and characterization of the MRIN
The MRIN was engineered by our developed ultra-pH-sensitive (UPS) nanotechnology[27–29]. The UPS polymers were firstly synthesized by atom transfer radical polymerization (ATRP) method, and then conjugated with Cy5-NHS and Cy7.5-NHS esters to obtain UPS-Cy5 and UPS-Cy7.5 fluorescent polymers, respectively (Supplementary Fig. 1). To fabricate a pH/light dual-responsive MRIN, an Always-ON micelle with consistent Cy5 signals either at micelle or dissociated states was prepared by blending 5% of UPS-Cy5 polymer (wt./wt.) with 95% of dye-free polymer (Supplementary Fig. 2a). Then, the pH/light dual-responsive monochromatic ratiometric imaging nanoparticle (MRIN) was constructed by replacing dye-free polymer (45%) in the Always-ON micelle with same amount of UPS-Cy7.5 polymer (Supplementary Fig. 2b, Supplementary Table 1). In MRIN, Cy7.5 served as a fluorescence quencher of Cy5 through FRET effect. At physiological pH, MRIN was micelle state with Cy5 signal "OFF", and the Cy5 signals can be fully activated through two mechanisms: pH-induced micelles dissociation (intracellular mechanism) and 808 nm irradiation-induced Cy7.5 photobleaching (extracellular mechanism) (Fig. 2a, b and Supplementary Fig. 2c–h). MRIN exhibited a sharp pH response ($\Delta pH_{10-90\%}$ = 0.19) with a transition pH (pH$_t$) of 6.26 and a high Cy5 signal activation ratio (110-fold), enabling the accurately reporting of nanoparticle internalization (Fig. 2c and Supplementary Fig. 2f). On one hand, upon addition of HCl, MRIN can quickly dissociate into unimers with Cy5 signal fully recovered (Fig. 2d). On the other hand, upon exposure to 808 nm laser, the Cy5 signal of MRIN was fully recovered along with complete photobleaching of Cy7.5 signal within 4 min (Fig. 2e and Supplementary Fig. 2g, h). More importantly, the Cy5 signals recovered through the above two mechanisms exhibited almost the same Cy5 fluorescence at each concentration of MRIN by fluorimeter and fluorescence imaging instrument (Fig. 2f), which is an important prerequisite for quantitative analysis of the intracellular and extracellular distribution of MRIN. The non-crosstalk activation of Cy5 signals by pH and laser irradiation was further verified by the fluorescence imaging analyses (Fig. 2g). Besides, MRIN exhibited excellent stability in fresh mouse plasma at 37 °C within 24 h (Supplementary Fig. 2i). The 'turn-on' mechanisms of MRIN based on pH-induced dissociation and 808 nm laser irradiation are shown in Supplementary Fig. 3.

### Methodology of MRIN for dissecting nanoparticle microdistribution
To demonstrate the feasibility of MRIN for quantitative imaging of nanoparticle microdistribution in vitro and in vivo, we established Cy5-labeled pH-insensitive micelle (RNP$_{0\%}$) and Always-ON micelle (RNP$_{100\%}$) as 0% and 100% endocytosis reference nanoparticles, respectively. The Cy5 signal of RNP$_{0\%}$ was completely 'OFF' whether cellular endocytosis or not due to its pH-insensitive nanostructure, and only fluorescently activated after 808 nm irradiation. In contrast, the Cy5 signal of RNP$_{100\%}$ was completely 'ON' whether endocytosis into the cells or not due to the Always-ON design, and not affected by 808 nm laser irradiation. For MRIN, the Cy5 signal of which endocytosed into the cells was fully activated (Supplementary Fig. 4a), whereas the Cy5 signal of extracellular nanoparticle was 'OFF' and can be switched 'ON' by 808 nm irradiation subsequently (Fig. 3a). The fluorescence behavior of three nanoparticles has been demonstrated in 4T1 breast cancer cell line. As shown in Fig. 3b, the Cy5 signal of RNP$_{0\%}$ cannot be detected in the extracellular and intracellular regions after

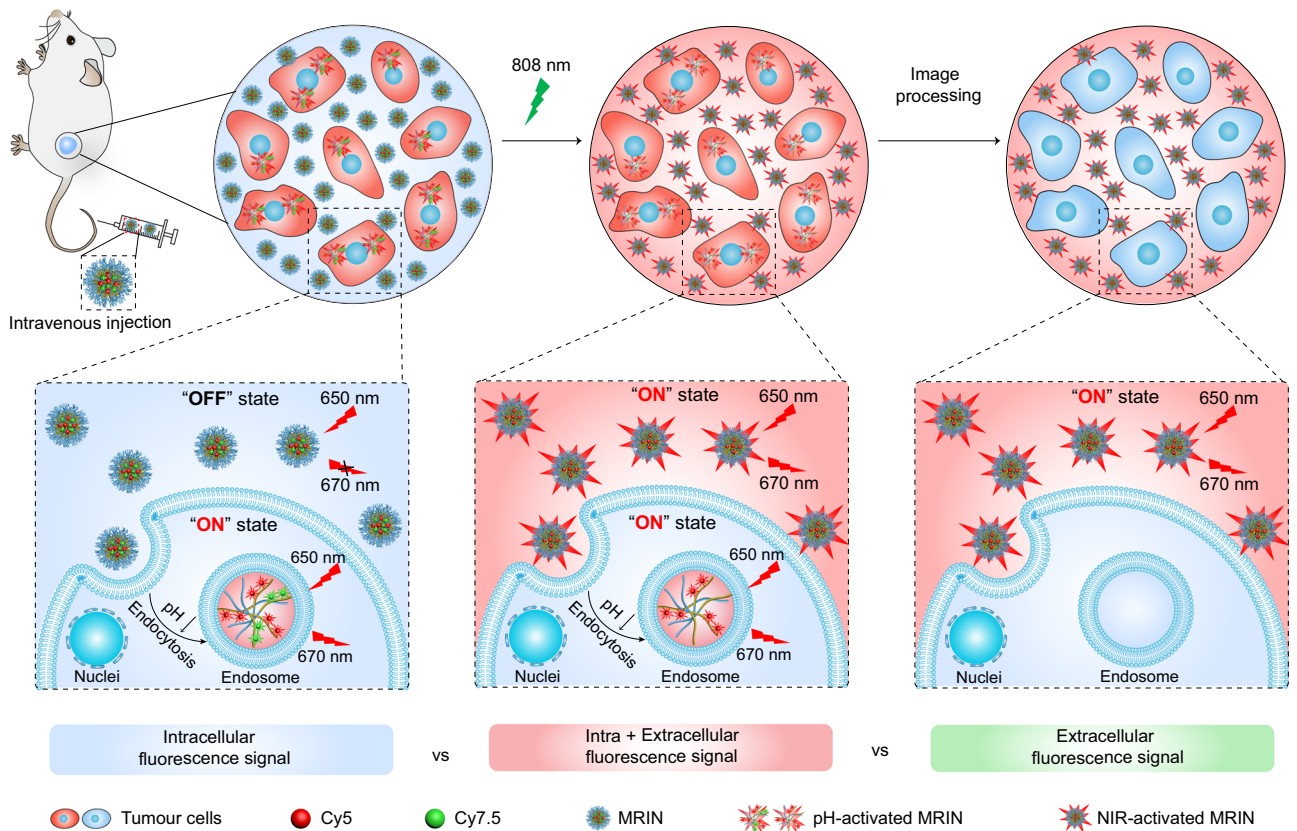

**Fig. 1 Schematic illustration of the monochromatic ratiometric imaging for quantifying extracellular and intracellular distribution of nanoparticles in living mice.** When accumulating at tumor tissues after intravenous injection, a part of MRINs are endocytosed into cells, then the Cy5 signals of intracellular nanoparticles are activated by the acidic pH of endosome. While the Cy5 signals of extracellular nanoparticles still keep 'OFF' state, which could be subsequently activated by 808 nm irradiation-induced Cy7.5 photobleaching. Harnessing MRIN nanotechnology, we can accurately quantify the intracellular and total (intracellular + extracellular) exposure of nanoparticles in tumor sites, thereby quantify the extracellular exposure of nanoparticles in tumor mass.

nanoparticle treatment, whereas the fluorescent signal was completely activated after 808 nm photo-irradiation. Reversely, the Cy5 signal of RNP$_{100\%}$ can be observed throughout the cell medium and intracellular compartments, and the signals kept constant after 808 nm laser irradiation. Using ImageJ software, the images of extracellular nanoparticles were obtained by subtracting images of non-irradiation from those of laser irradiation. We can find an excellent black and white pattern images of intracellular and extracellular nanoparticles for RNP$_{0\%}$, as well as a reversed pattern for RNP$_{100\%}$. These results demonstrated that RNP$_{0\%}$ and RNP$_{100\%}$ are suitable to simulate the artificial states of 0% and 100% endocytosis regardless of their real distribution in extracellular and intracellular compartments. For the cells incubated with MRIN, punctate Cy5 signals were detected in acidic endocytic organelles (pseudo-colored green), and a large portion of extracellular nanoparticle signals (pseudo-colored red) were lighted up post-irradiation. The intracellular and extracellular distributed nanoparticles were successfully differentiated in the merged images by non-crosstalk monochromatic imaging. The ratiometric images of intracellular and extracellular distribution of nanoparticles can be easily calculated by ImageJ software (Fig. 3b). Furthermore, the intracellular fluorescence behavior of three nanoparticles was also studied by flow cytometry (Supplementary Fig. 4b). The intracellular Cy5 signal of RNP$_{0\%}$ was completely activated after 808 nm irradiation with over 40-fold signal amplification. Whereas, the intracellular Cy5 signal of RNP$_{100\%}$ kept constant after 808 nm laser irradiation due to the Always-ON design. For MRIN, negligible enhancement of

intracellular Cy5 signal was observed after 808 nm laser irradiation, demonstrating that the internalized MRIN were completely activated in the endo-lysosomes.

To perform the in vivo imaging experiment, the 808 nm irradiation time on tumors for Cy5 signal activation was optimized. The Cy5 fluorescence of irradiated tumor rapidly increased along with the photobleaching of Cy7.5 signals and reached plateau within 10 min, while the Cy5 fluorescence of unirradiated tumor remained unchanged (Supplementary Fig. 5a, b). Therefore, 10 min was selected as the optimal irradiation time for further study in vivo. Given that 808 nm irradiation has been reported to be widely used for photothermal therapy of tumors[30–33], we next evaluated the tumor tissue damage triggered by near-infrared (NIR) irradiation. Our results revealed that marginal vascular damage and tumor apoptosis were observed after NIR treatment below 42 °C, which was almost the same as PBS negative control. However, the 52 °C treatment group as a positive control exhibited notable vascular destruction and tumor apoptosis (Supplementary Fig. 5c–f). Therefore, irradiation with 808 nm laser at 42 °C for 10 min only induces a mild hyperthermia without tumor photodamage, which was selected for NIR photobleaching of Cy7.5 in vivo.

We then evaluated the accuracy of our quantitative method using RNP$_{0\%}$ and RNP$_{100\%}$ as reference standards of 0% and 100% endocytosis in vivo, respectively. The nanoparticles were intravenously injected into 4T1 bilateral tumor bearing mice. As shown in Fig. 3c, the Cy5 signals of left and right tumors can be hardly detected before 808 nm laser irradiation in RNP$_{0\%}$-treated mice, while the fluorescent readout of right tumor was

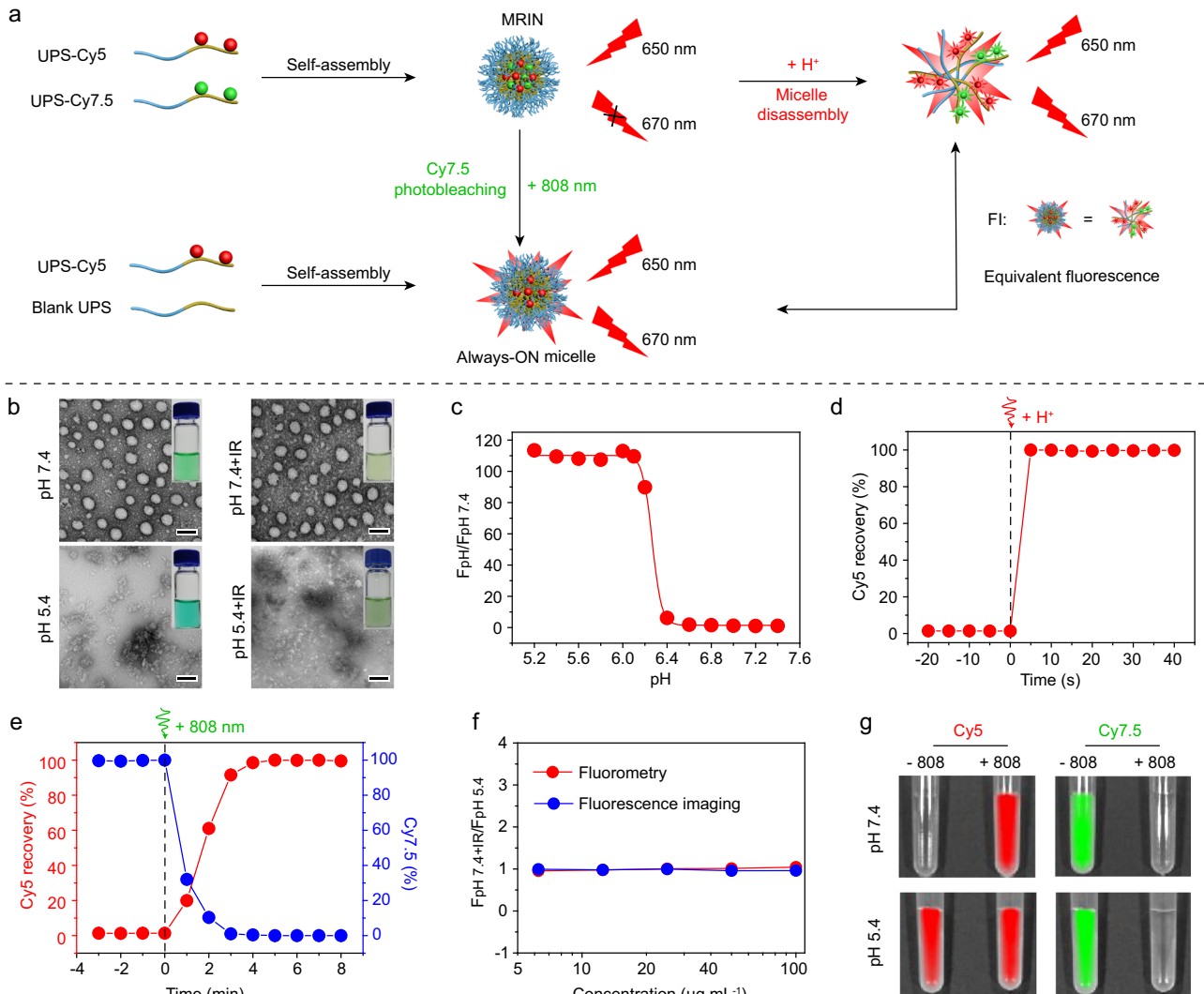

**Fig. 2 pH and light sensitivity of MRIN. a** Schematic illustration of pH- and light-triggered Cy5 fluorescence recovery mechanisms. **b** The photographic and TEM images of MRIN at pH 7.4 and pH 5.4 with or without 808 nm irradiation. Scale bar = 50 nm. **c** Cy5 fluorescence ratios as a function of pH values for MRIN at 37 °C. **d** pH-triggered Cy5 fluorescence recovery versus time profile upon hydrochloric acid (HCl) addition. **e** Light-triggered Cy5 fluorescence recovery and Cy7.5 photobleaching of MRIN. The percentage of Cy5 fluorescence recovery and Cy7.5 fluorescence decay versus time profile upon 808 nm irradiation (0.5 W cm$^{-2}$). **f** The comparison of Cy5 fluorescence recovery by pH-induced micelles dissociation and light-induced Cy7.5 photobleaching. The Cy5 fluorescence signals were recorded using fluorescence spectrophotometer or IVIS in vivo imaging system. **g** Fluorescence images of MRIN in pH 7.4 and pH 5.4 PBS buffers with or without 808 nm irradiation.

exponentially enhanced after 808 nm irradiation. In a reverse pattern, strong Cy5 signals of RNP$_{100\%}$ were captured in left and right tumors before laser irradiation, and the signals kept constant before and after irradiation. As for MRIN, the Cy5 signals of both side tumors were partially activated, and the fluorescent readouts of right tumors were significantly enhanced after irradiation. The increased Cy5 signals at the tumor sites can be ascribed to the contribution of extracellular nanoparticles. Thus, the extracellular Cy5 fluorescence images can be easily calculated by subtracting Cy5 fluorescence of pre-irradiated tumors from that in post-irradiated same ones. The extracellular percentages of RNP$_{0\%}$, MRIN, and RNP$_{100\%}$ were quantitatively analyzed to be 94.9%, 75.8%, and 2.1%, respectively (Fig. 3d). Accordingly, the endocytic percentage of RNP$_{0\%}$, MRIN, and RNP$_{100\%}$ were 5.1%, 24.2%, and 97.9%, respectively. The simulated extracellular distribution of RNP$_{0\%}$ and RNP$_{100\%}$ kept 93.4–94.9% and −0.2–2.1% over 24 h in 4T1-tumor bearing mice (Supplementary Fig. 6a, b), demonstrating the robustness of

RNP$_{0\%}$ and RNP$_{100\%}$ as the reference standards of nanoparticle microdistribution. It's worth noting that although both RNP$_{100\%}$ (Always-ON micelle) and MRIN exhibited good tumor targeting effect as a result of the enhanced permeability and retention (EPR) effect, MRIN achieved significantly higher tumor-to-normal tissue contrast due to the pH-responsive Cy5 signal amplification (Supplementary Fig. 6c). The fluorescence imaging of tumor slices was also conducted for more in-depth validation (Fig. 3e and Supplementary Fig. 7). There were marginal NPs endocytosed into the cells in the RNP$_{0\%}$ group, and almost all NPs in RNP$_{100\%}$ group were endocytosed, while a small portion of NPs in MRIN group were endocytosed by the cells (~24%). The extracellular and intracellular distributions of MRIN in tumor area were presented in a black and white manner (Fig. 3f). These results were consistent with those at cellular level and animal models. Collectively, our results demonstrated that the intracellular and extracellular distribution of MRIN can be accurately quantified by monochromatic imaging.

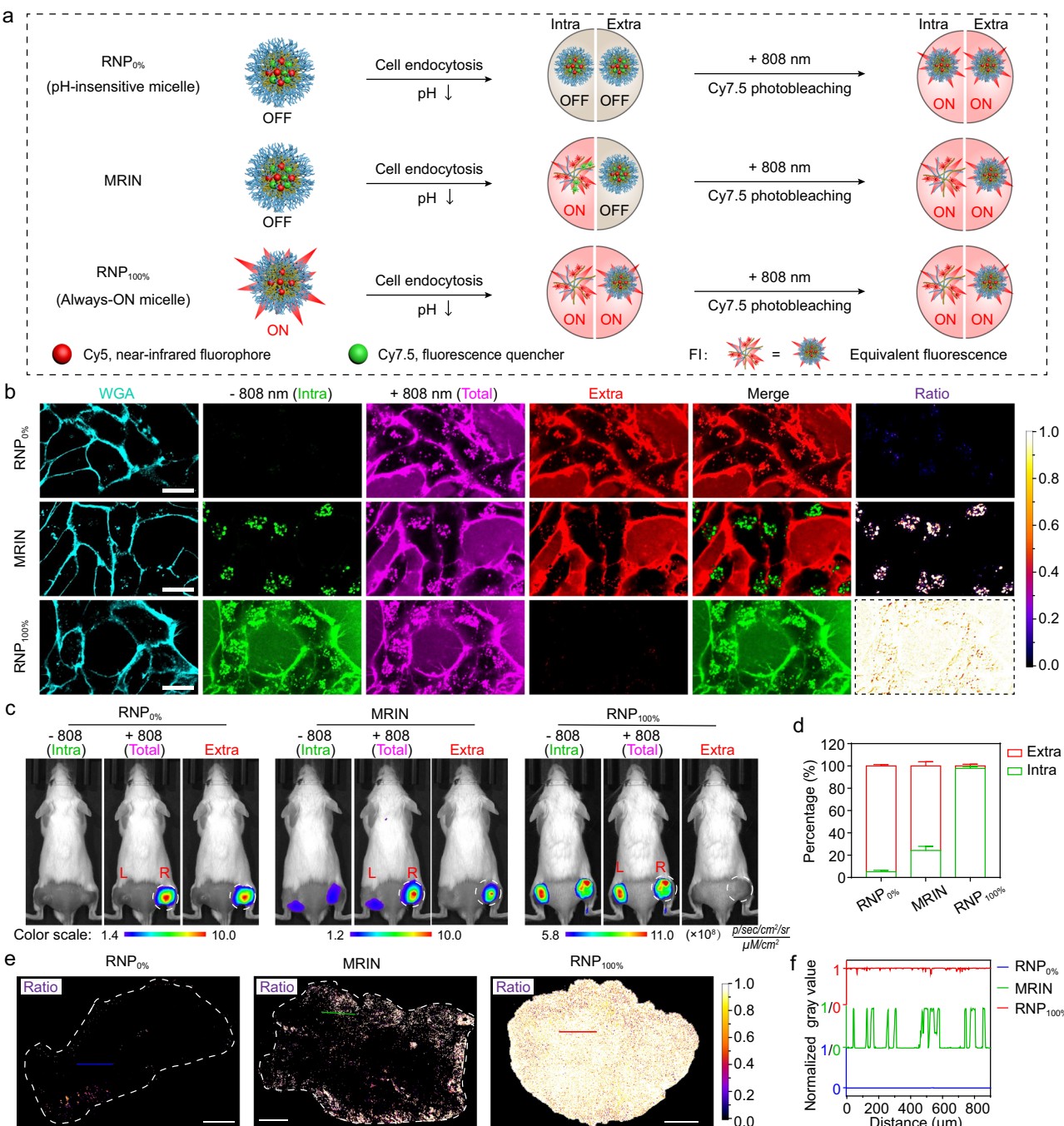

**Fig. 3 Methodology of MRIN for quantifying intracellular and extracellular distribution of nanoparticles. a** Schematic illustration of Cy5 fluorescence amplification for $RNP_{0\%}$, MRIN, and $RNP_{100\%}$ in response to sequential cellular endocytosis and 808 nm irradiation. **b** 4T1 cells were incubated with $RNP_{0\%}$, MRIN, and $RNP_{100\%}$ at 37 °C for 0.5 h, respectively, then the cells were irradiated in situ with an 808 nm laser. The image of extracellular nanoparticles was calculated by subtracting the pre-irradiated Cy5 fluorescence image from the post-irradiated Cy5 fluorescence image, and the merged images were presented in pseudo-color green and red for intracellular and extracellular NPs, respectively. The ratio channel displayed the intracellular distribution of nanoparticles. The images of $RNP_{0\%}$ and $RNP_{100\%}$ before 808 nm irradiation were served as the extrapolated states of 0% and 100% internalization of nanoparticles. Scale bar = 20 μm. **c** In vivo fluorescence images of the bilateral 4T1 tumor-bearing mice with irradiation on right tumors at 24 h post-injection of $RNP_{0\%}$, MRIN, and $RNP_{100\%}$. The images of extracellular nanoparticles were obtained by subtracting pre-irradiated Cy5 fluorescence images from post-irradiated Cy5 fluorescence images. The circles indicate the irradiated right tumors. **d** The percentage of intracellular and extracellular nanoparticle exposure from Fig. 3c ($n = 4$ biologically independent mice). **e** The ratiometric fluorescence images of intracellular nanoparticles versus total distributed nanoparticles in tumor sections at 24 h post-injection. **f** The values of ratiometric signal indicate endocytic events cross the corresponding line in Fig. 3e, value "1" and "0" represent endocytic NPs and non-endocytic NPs, respectively. All data were presented as mean ± s.d.

To further investigate the accuracy of the quantitative method, we performed the following experiments. Firstly, the 808 nm irradiation selectively photobleached Cy7.5, while had no influence on Cy5 signal (Supplementary Fig. 8a). Secondly, after irradiation with 808 nm laser, the Cy7.5 of the UPS-Cy7.5 micelle and Cy5/Cy7.5 hybrid micelle (MRIN) were completely photobleached in vitro. As a result, the Cy5 signal of MRIN was fully recovered. Moreover, the by-product from Cy7.5 photobleaching showed no fluorescence signal in Cy5 channel (Supplementary Fig. 8b). The same phenomenon was verified in the mice bearing bilateral 4T1 tumor model (Supplementary Fig. 8c). In addition, we investigated whether the mild hyperthermia induced by 808 nm laser irradiation would increase the nanoparticle accumulation in tumor sites, which may cause inaccurate quantification. The mice bearing bilateral 4T1 tumors were intravenously injected with cocktail of UPS-Cy5 and UPS-Cy7.5 micelles. The right tumors were irradiated with 808 nm laser at 24 h post-injection and the tumor temperature was maintained at about 42 °C. As seen in Supplementary Fig. 8d, the Cy7.5 fluorescence of right tumor was photobleached after irradiation, while no change in Cy5 fluorescence was found before and after photobleaching. Thus, the nanoparticle accumulation caused by mild hyperthermia can be ignored during a very short time interval after NIR irradiation (<15 min). The imaging also demonstrated that the cocktail of UPS-Cy5 and UPS-Cy7.5 micelles failed to achieve the ON/OFF switch of Cy5 signal due to the large distance between Cy5 and Cy7.5 molecules.

**Quantitative imaging of extracellular and intracellular distribution of nanoparticles in different tumors**. Having demonstrated the accuracy of the quantitative method, we next harnessed the MRIN to accurately quantify extracellular and intracellular distribution of nanoparticles in different tumor models. Firstly, the microdistribution of nanoparticles in 4T1 tumor-bearing mice was quantified over 4 days. As shown in Fig. 4a, the Cy5 signals of tumors were greatly increased along with the disappearance of Cy7.5 signals at each time point after irradiation. After data processing, the fluorescence images of extracellular nanoparticles were successfully calculated. The quantitative results revealed that the exposure level of nanoparticles in intracellular and extracellular compartments of tumor tissues was continuously enhanced within 48 h post-injection. However, the intracellular nanoparticle exposure remained unchanged, while the extracellular nanoparticle exposure was gradually decreased in the following 2 days, which probably due to the slow clearance of nanoparticles from mice (Fig. 4b). Accordingly, the percentage of extracellular nanoparticle exposure and the ratio of extracellular versus intracellular exposure reached a maximum at 48 h post-injection followed by a slight decrease in the subsequent monitoring period. The percentages of extracellularly distributed nanoparticles accounted for 70-80% of total accumulated ones during 96 h post-injection (Fig. 4c). The slightly decreased intracellular percentage from 3 h to 12 h is owing to enhanced total tumor accumulation over time and the exhausting of the bio-nano interaction receptors/proteins on the cell membrane[8,34], so that impair additional receptor-mediated endocytosis of nanoparticles. The non-crosstalk imaging of intracellularly and extracellularly distributed nanoparticles in the tumor sections was also achieved by pH/light sequentially activated monochromatic imaging (Fig. 4d). The quantitative imaging analyses of intracellularly and extracellularly distributed nanoparticles in five tumor-bearing mice models, including orthotopic breast cancer and four subcutaneous cancer models (breast, colon, pancreatic cancers), were also performed. The ratiometric images of extracellular signals versus total

accumulated ones at 24 h post-administration were generated, demonstrating the heterogeneity of extracellular distribution of NPs in different tumor types spanning from 65% to 80% (Fig. 4e and Supplementary Fig. 9). Moreover, heat map indicated the heterogeneity of extracellular distribution of NPs within the same tumor types (Fig. 4f). The tumor accumulation and extracellular distribution of NPs increased gradually in all five tumor models over 24 h (Fig. 4g and Supplementary Fig. 10). For most tumors, about 50-80% of the total accumulated MRIN was distributed in the extracellular space of tumor mass over 24 h post-administration, indicating that the majority of NPs are trapped in acellular tumor stroma.

**Parsing microdistribution of MRITN on photodynamic therapeutic contribution**. Currently, activatable nanoparticles for photodynamic therapy (PDT) mostly rely on the intracellular exposure to exert lethal tumor damage, while a large portion of extracellular particles stay silent and useless[35]. Therefore, it requires a high photosensitizer dose or high irradiation power to achieve the desired therapeutic effect[36,37]. In this study, a PDT-based MRITN was constructed to parse extracellular and intracellular distribution of nanoparticles on therapeutic contribution, thereby significantly improve the therapeutic efficacy through combined extracellular and intracellular PDT treatments.

The MRITN was prepared by blending of UPS-Ce6 polymer with the same amount of UPS-Cy7.5 polymer (Supplementary Table 1). Cy7.5 could quench the fluorescence and photosensitivity of Ce6 due to the FRET effect between them (Supplementary Fig. 11). Therefore, MRITN could achieve pH and light activatable Ce6 fluorescence recovery and singlet oxygen generation (SOG) (Supplementary Fig. 12). Although Cy7.5 can also work as photosensitizers, its singlet oxygen generation under 808 nm irradiation was 20-fold lower than MRITN under 660 nm irradiation. Hence, compared with Ce6, the SOG of Cy7.5 was negligible for the photodynamic effect of MRITN (Supplementary Fig. 13). At cellular level, we could regulate the location of MRITN to achieve spatially controlled activation of Ce6 and singlet oxygen. The cytotoxic singlet oxygen could be controlled to generate only inside the cells, only outside the cells, or both inside and outside the cells (Fig. 5a). Based on the controllable spatial activation of SOG, the therapeutic efficacy of intracellular and extracellular PDT was next investigated, respectively. As shown in Fig. 5b, the IC$_{50}$ of intracellular PDT group and extracellular PDT group were 8.25 µg mL$^{-1}$ and 6.82 µg mL$^{-1}$, respectively. Damage to the cell membranes plays an important contribution to the extracellular PDT. We found that MRITN could bind to the cell membrane at physiological pH, and the cell binding ability of MRITN was significantly enhanced in a slightly acidic environment such as tumor microenvironment (Supplementary Fig. 14). Surprisingly, the combined intracellular and extracellular PDT group significantly reduced the IC$_{50}$ of Ce6 to 2.08 µg mL$^{-1}$. We performed the cell apoptosis experiments to further investigate the efficacy of combined PDT. The results were consistent with the cytotoxicity test, that is the combined PDT group exhibited the most efficacious cell apoptosis as compared with other groups (Fig. 5c, d).

For in vivo imaging, the MRITN could also achieve precisely quantitative imaging of endocytosed and extracellular nanoparticles in tumor tissues (Supplementary Fig. 15a, b). Pharmacokinetics experiment showed that MRITN exhibited a long circulation time with an elimination half-life of 18.8 h (Supplementary Fig. 15c, d). In the antitumour study, the 4T1 tumor-bearing mice were treated with MRITN at a Ce6 concentration of 0.75 mg kg$^{-1}$, which was 6-fold lower than most reported studies for PDT therapy[38,39]. The laser irradiation was performed at 3 h or 24 h

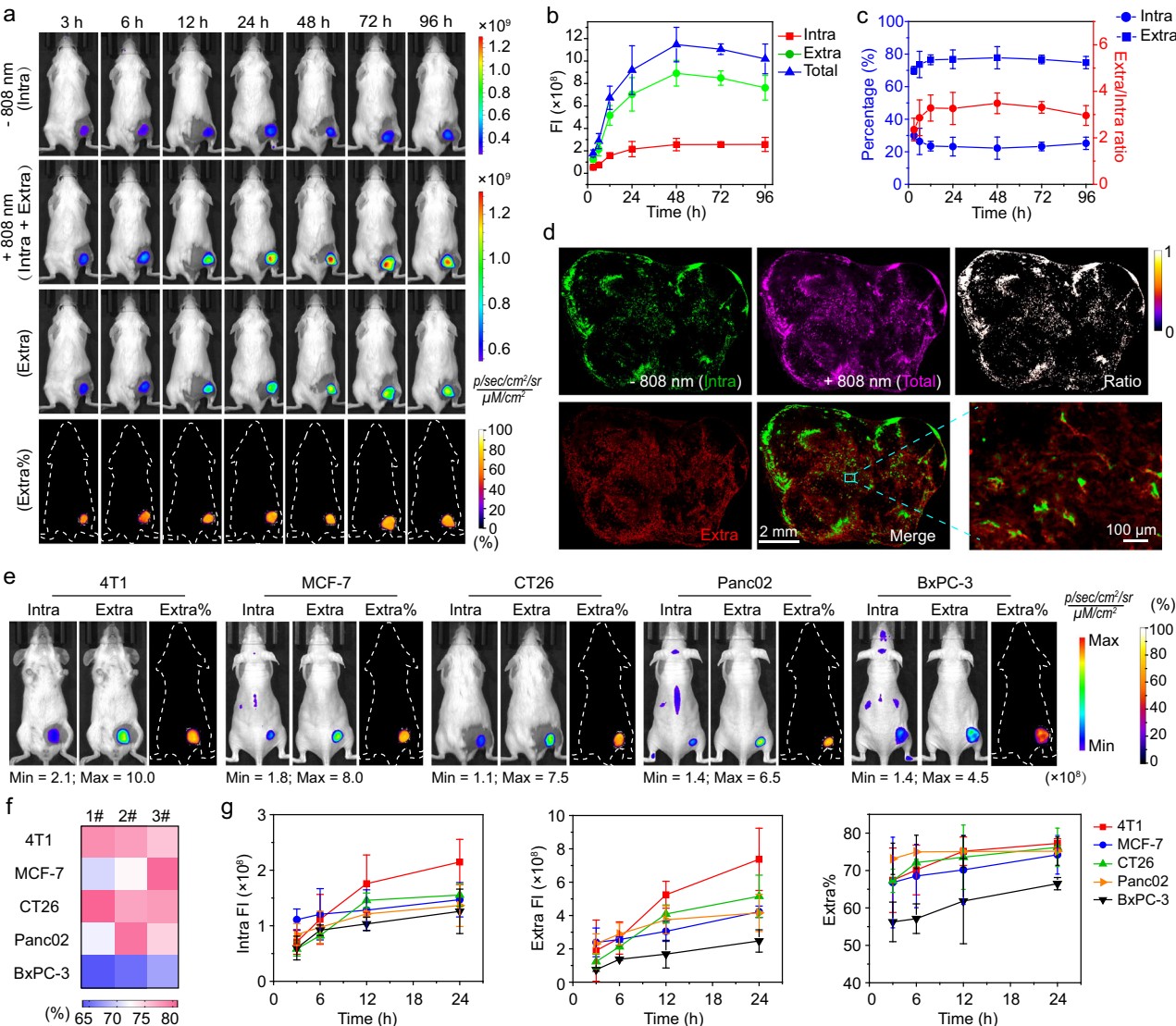

**Fig. 4 Quantitative imaging of extracellular and intracellular distribution of nanoparticles in different tumors. a** The pre- and post-irradiated Cy5 fluorescence images, calculated extracellular Cy5 fluorescence images, and the ratiometric images of extracellular signals versus post-irradiated fluorescence signals of 4T1 tumor-bearing mice at different time points post-injection of MRIN. **b** The time-dependent profile of intracellular, extracellular, and total nanoparticle exposure in 4T1 tumor sites over 96 h ($n = 4$ biologically independent mice). **c** The percentages of intracellular and extracellular distributions of MRIN as well as the ratio of extracellular exposure to intracellular ones in 4T1 tumor sites over 96 h ($n = 4$ biologically independent mice). **d** The intracellular, extracellular, total distribution of nanoparticles in tumor sections at 24 h post-injection of MRIN. The merged image showed the unambiguous differentiation of extracellular and intracellular nanoparticles. The ratiometric images of intracellular exposure of nanoparticles were also calculated. **e** Dissecting the intracellular and extracellular distribution of NPs in different tumors at 24 h post-injection of MRIN. **f** Heat map shows the percentages of extracellular nanoparticles in various tumor models at 24 h post-injection of MRIN ($n = 3$ biologically independent mice). **g** The quantitative intracellular and extracellular nanoparticle exposure in different tumors at different time post-injection ($n = 3$ biologically independent mice). The percentages of extracellular nanoparticles ranged from 50-80% of total accumulated NPs over 24 h. All data were presented as mean ± s.d.

post-injection of MRITN in two separate experiments. Firstly, the antitumour efficacy of MRITN with a drug-light interval (DLI) of 3 h was evaluated. As shown in Fig. 5e, the antitumour efficacy of MRITN with or without 808 nm laser irradiation groups has no significant difference as compared with PBS group over a time course of 19 days, demonstrating that 808 nm laser treatment alone had marginal inhibition on tumor growth. For MRITN combined with 660 nm laser group (intracellular PDT) and MRITN combined with 660 nm + 808 nm group, the antitumour efficacy was significantly enhanced as compared with MRITN plus 808 nm group. Notably, the MRITN combined with 660 nm + 808 nm group did not improve the PDT efficacy as compared with MRITN combined with 660 nm group, indicating that the 808 nm

irradiation only played a marginal role in antitumour efficacy when applied after 660 nm irradiation. However, harnessing 808 nm irradiation before 660 nm (intracellular + extracellular PDT) resulted in the most efficient antitumour efficacy (Fig. 5f) and the longest lifespan as compared with the other five groups (Fig. 5g). The tumor growth inhibition was dramatically enhanced from 52.2% for intracellular PDT to 94.5% for the combined PDT. This superior antitumour efficacy is because that the extracellular Ce6 was activated after 808 nm irradiation, and then singlet oxygen generated both inside and outside the cells under 660 nm irradiation, thereby realizing combined intracellular and extracellular PDT. In addition, the antitumour study of MRITN with DLI of 24 h was also performed. Similarly, combined PDT

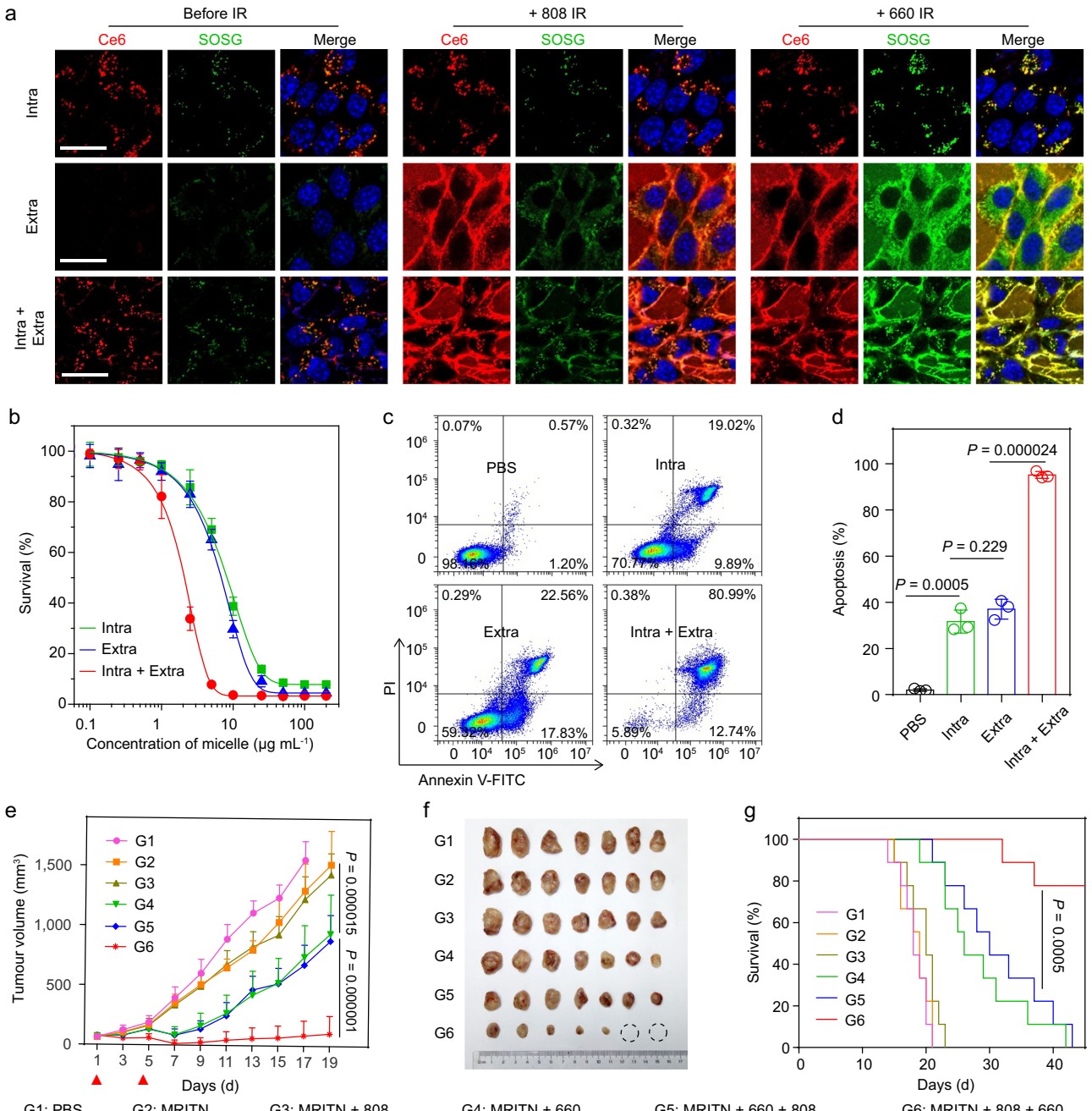

**Fig. 5 Combined extracellular and intracellular photodynamic therapy for improved therapeutic outcome. a** Spatially controllable Ce6/SOG activation of MRITN. Top panel: intracellular Ce6/SOG activation. Cells were incubated with MRITN at 37 °C, followed by the removal of extracellular MRITN. Middle panel: extracellular Ce6/SOG activation. Bottom panel: Combined intracellular and extracellular Ce6/SOG activation. Cells were incubated with MRITN at 4 °C (Middle panel) and 37 °C (Bottom panel) without removal of extracellular MRITN. Scale bar =20 μm. **b** In vitro cytotoxicity of MRITN with spatially controllable PDT activation. 4T1 cells were irradiated with a 660 nm laser (100 mW cm$^{-2}$ for 3 min) at 4 °C under dark conditions ($n=3$ biologically independent cell samples). **c** Flow cytometry analysis of 4T1 cells apoptosis under 660 nm irradiation by spatially controllable PDT therapy. **d** Quantitative apoptotic percentage of 4T1 cells based on flow cytometry ($n=3$ biologically independent cell samples). Statistical analysis by two-sided Student $t$-test. **e** Tumor growth curves in subcutaneous 4T1 tumor-bearing mice with different irradiation treatments. Mice were irradiated with 660 nm laser at 400 mW cm$^{-2}$ for 10 min at 3 h post-injection of MRITN (Ce6 dose of 0.75 mg kg$^{-1}$). The PDT treatment was given twice separately at day 1 and day 5 ($n=8$ biologically independent mice). Two-way analysis of variance (ANOVA). **f** Photographic images of excised tumors in different treatment groups at the end of antitumour study. **g** The survival rates of mice bearing 4T1 tumors after different irradiation treatments ($n=8$ biologically independent mice). Survival analysis was performed by the log-rank (Mantel–Cox) test. All data are presented as mean ± s.d.

achieved the most effective tumor inhibition as compared with other treatment groups (Supplementary Fig. 16). The H&E staining and TUNEL analysis revealed that the MRITN exhibited good biocompatibility without distinct toxicity to the main organs (Supplementary Fig. 17).

**Combined PDT inhibited tumor metastasis by remodeling the tumor microenvironment**. In accordance with previous studies[40,41], we found that the combined PDT could remodel the tumor microenvironment by significantly reducing the amounts of cancer associated fibroblasts (CAFs) and the density of extracellular matrix proteins, such as collagen and fibronectin (Supplementary Fig. 18). It has been demonstrated that extracellular matrix plays an important role in assisting tumor metastasis[42]. Tumor stroma is characterized by extracellular matrix (ECM) stiffening. The stiff ECM promotes tumor cell growth, invasion, and migration through the formation of the cross-linked collagen tracks[43,44]. Therefore, we next investigated the anti-metastasis effect of combined PDT therapy. The 4T1-luciferase tumor-bearing mice model was established through subcutaneous inoculation of tumor cells into the right flanks. The tumor-bearing mice received the MRITN-mediated PDT treatment at 6 days after the tumor implantation. Then, the lung metastasis was evaluated by luminescence imaging every three days (Fig. 6a). As presented in Fig. 6b, c, MRITN + 808 + 660 group showed significant antitumour metastasis as compared to the single intracellular PDT (MRITN + 660 group), which exhibited a marginal inhibition of lung metastasis. The inhibition of metastasis by combined intracellular and extracellular PDT was further verified by H&E staining of the lung tissues (Fig. 6d).

Finally, we tried to understand the potential mechanism for the anti-metastasis effect of combined PDT. The cell-ECM interactions are mainly mediated by integrin β1, a type of adhesion molecules located in cell membranes[45]. It has been reported that the downregulation of integrin β1 and ECM destruction both contribute to the anoikis (a kind of programmed cell death induced by cell detachment from ECM) of tumor cells and inhibit tumor metastasis[46,47]. Our study found that the mild extracellular PDT (>85% cell survival) could dramatically downregulate the expression of integrin β1 in 4T1 and MCF-7 two breast cancer cell lines as compared with intracellular PDT (Fig. 6e–g), indicating that the extracellular PDT could inhibit tumor metastasis by inducing anoikis of tumor cells. Overall, compared with intracellular PDT alone, combined PDT synergistically enhanced the antitumour efficacy with inhibited tumor metastasis through the intracellular PDT mediated cell killing together with anoikis of tumor cells induced by integrin β1 downregulation and ECM destruction. However, the tumor metastasis is a very complicated process, the ECM destruction and integrin β1 downregulation could be two of the most important factors for anti-metastasis. A more-in-depth mechanism needs to be investigated in the future.

## Discussions
We developed a pH/light dual-responsive monochromatic ratiometric imaging nanotechnology to dissect extracellular and intracellular distribution of nanoparticles in tumor tissues. The ultra-pH-sensitivity and exponential signal amplification of MRIN enable its unequivocal differentiation between extracellular and intracellular nanoparticles in living animals. At tumor sites, the fluorescence signals of intracellular MRIN were 'ON' due to endocytic pH-induced signal amplification. In contrast, the fluorescence signals of extracellular MRIN were 'OFF', which can be turned 'ON' through photoirradiation-induced signal amplification. Benefiting from the unique property of MRIN that can

selectively control the ON/OFF states of intracellular and extracellular signals, we can easily quantify the intracellularly and extracellularly distributed nanoparticles in tumor mass without signal crosstalk. Compared with classic two-channel ratiometric probes[48,49], MRIN overcomes the problems such as the interference between two channels and different penetration depths in vivo, enabling its precise quantification. Harnessing $RNP_{0\%}$ and $RNP_{100\%}$ as extrapolated states for 0% and 100% endocytosis in vivo, we validated the monochromatic quantitative methodology. Enabled by MRIN, the intracellular and extracellular distributions of nanoparticles were successfully imaged in different tumor models, which greatly contributes to the deep understanding and prediction of intracellular drug delivery efficacy and therapeutic outcome.

For most nanomedicines, the therapeutic efficacy greatly depends on their cellular internalization efficiency, while the extracellular NPs are wasted. Inspired by the quantitative results that numerous NPs reside extracellularly, a MRITN for photodynamic therapy was developed to parse the extracellularly and intracellularly distributed nanoparticles on therapeutic contribution. Currently, photodynamic therapy (PDT) has been widely exploited for the treatment of various malignant tumors due to its minimal invasion and fast healing. The responses to PDT rely largely on the light dose, photosensitizer concentration and location[50]. In addition to the vascular damage and direct cell killing effect at therapeutic dose level, photodynamic priming (PDP) using subtherapeutic dose has also been demonstrated to efficiently enhance the antitumour efficacy of subsequent therapies. Several researches have revealed that PDP can sensitize tumors to immuno- and chemo-therapy via the tumor microenvironment modulation, including the decrease of extracellular matrix content and the enhancement of tumor vascular leakiness[51,52]. In this study, MRITN can selectively achieve intracellular PDT or combined intracellular and extracellular PDT. Our results demonstrated that the extracellular NPs played an equal important role in cancer treatment via both cell membrane damage (direct cell killing) and ECM destruction (photodynamic priming), enabling the maximized therapeutic outcomes of combined PDT. Moreover, the combined PDT promoted tumor anoikis and significantly reduced the lung metastasis. Based on our findings, we speculate that the potential mechanism for the anti-metastasis effect of combined PDT is the ECM destruction and the downregulation of adhesion integrin β1. However, the tumor metastasis is a very complicated process, and influenced by various physiological factors. Although previous investigations have reported that excessive reduction of extracellular matrix and stromal cells promotes tumor metastasis[53,54], our MRITN-mediated PDT achieved remarkable inhibition of lung metastasis probably due to the lethal damage to tumor cells. However, the comprehensive mechanism needs to be further investigated in the future.

In summary, we successfully quantified the intracellular and extracellular distribution of nanoparticles at cellular, tissue, and whole animal levels without any crosstalk by our MRIN technology. This nanotechnology offers a powerful tool in parsing the contribution of intracellularly and extracellularly distributed nanomedicines to therapeutic outcomes.

## Methods
**Materials**. Chlorin e6 (Ce6) was obtained from Frontier Scientific, Inc. (USA). Cy5 NHS and Cy7.5 NHS esters were purchased from Lumiprobe Company (Maryland, USA). The polymers including PEG$_{5k}$-$b$-poly(2-(ethylpropylamino) ethyl methacrylate-$r$-2-(dipropylamino) ethyl methacrylate-$r$-2-aminoethyl methacrylate) (PEG$_{5k}$-$b$-P(DPA$_{40}$-$r$-EPA$_{40}$-$r$-AMA$_3$)) and PEG$_{5k}$-$b$-poly(2-ethylhexyl methacrylate-$r$-2-aminoethyl methacrylate) (PEG$_{5k}$-$b$-PEH$_{80}$-$r$-AMA$_3$) were synthesized by our laboratory. Dicyclohexylcarbodiimide (DCC) and N-Hydroxysuccinimide (NHS) were purchased from Sigma-Aldrich. N,N-dimethyl-4-nitrosoaniline (RNO)

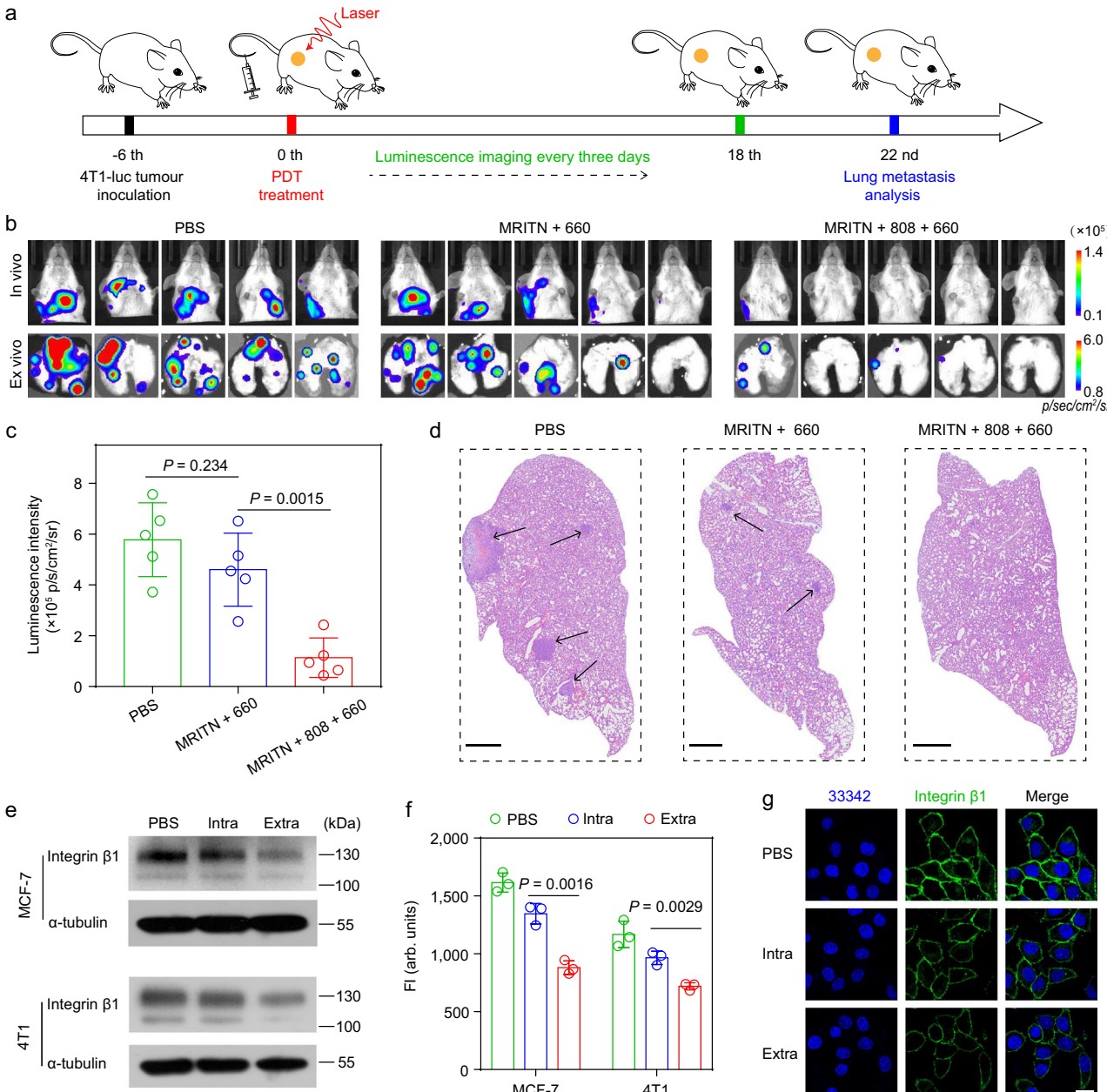

**Fig. 6 In vivo anti-metastasis effect by combined extracellular and intracellular PDT in 4T1 tumor-bearing mice. a** Therapeutic schedule for combined PDT. **b** In vivo and ex vivo lung metastasis of each group analyzed by bioluminescence imaging. **c** The quantitative result of ex vivo lung metastasis in Fig. 6b. Data were presented as mean ± s.d. ($n = 5$ biologically independent mice). Statistical analysis by two-sided Student t-test. **d** H&E staining of the lungs at the end of antitumour study. The areas pointed by the arrow were tumor metastasis nodules. Scale bar = 1 mm. **e–g** The downregulation of integrin β1 by intracellular PDT or extracellular PDT measured by **e** western blot, **f** flow cytometry ($n = 3$ biologically independent cell samples, two-sided Student t-test), and **g** confocal microscope. The tumor cells treated with intracellular PDT or extracellular PDT were washed with PBS and then lysed with RIPA lysis buffer for western blot analysis. The treated cells were stained with integrin β1 antibody followed by AF594-conjugated secondary antibody, then analyzed by flow cytometry and confocal microscope, respectively. Scale bar = 20 μm. All data were presented as mean ± s.d.

was purchased from Alfa Aesar (USA). 3-(4,5-dimethylthiazol-2-yl)-2,5-diphenyltetrazolium bromide (MTT) was purchased from Coolaber (China). Hoechst 33342, Singlet Oxygen Sensor Green (SOSG) and Wheat Germ Agglutinin-Alexa Fluor™ 488 conjugate (AF488-WGA) were obtained from Invitrogen Inc. (OR, USA). Fibroblast Marker (ER-TR7, 1:40) was purchased from Santa Cruz Biotechnology (Shanghai, China). Anti-CD31 antibody [MEC 7.46] (ab7388, 1:500), anti-integrin β1 antibodies (ab179471, 1:2000) and (ab24693, 1:200), AF594-conjugated Goat anti-Rat (ab150160, 1:1000), AF594-conjugated Goat anti-Mouse (ab96873, 1:1000), HRP-labeled Goat anti-Rabbit (ab6721, 1:5000) and Goat anti Mouse (ab6789, 1:10,000) secondary antibodies were purchased from Abcam (Shanghai, China). Anti-tubulin antibody (T5168, 1:10,000) was purchased from Sigma-Aldrich. Other solvents and reagents were all received from Sigma-Aldrich or Fisher Scientific Inc.

**Cells and animals**. The murine 4T1 breast carcinoma, CT26 colon carcinoma, and human MCF-7 breast carcinoma cell lines were obtained from the American Type Culture Collection (ATCC) and maintained in RPMI 1640 medium supplemented with 10% fetal bovine serum (FBS). Murine Panc02 and human BxPC-3 pancreatic cancer cell lines were purchased from National Infrastructure of Cell Line Resource and cultured in DMEM medium added with 10% FBS. The cells were incubated at 37 °C humidified environment with 5% $CO_2$ supply. Female nu/nu nude mice (18–21 g) were obtained from Vital River Laboratory Animal Center (Beijing, China) and Female BALB/c mice of 18–20 g were obtained from Peking University Health Science Center (Beijing, China). Animals were housed under SPF conditions in groups of 4–5 mice per cage, and maintained at a temperature of ~25 °C in a humidity-controlled environment with a 12 h light/dark cycle, with free access to standard food and water. To establish bilateral tumor models, the suspension of

tumor cells ($1 \times 10^6$ cells/tumor) were injected subcutaneously into the bilateral flanks of the mice. When the tumor volume reached approximately 50–100 mm³, the mice were used for the follow-up experiments such as in vivo fluorescence imaging and antitumour study. All animal procedures were carried out in accordance with the guidelines approved by the Institutional Animal Care and Use Committee (IACUC) of Peking University (Accreditation number: LA2021500). The maximum permitted tumor volume (2000 mm³) was not exceeded in any study. For monitoring tumor growth, mice were euthanized once the tumors had reached ~1500 mm³ in size.

**Syntheses of dye-conjugated functional polymers.** The ultra-pH-sensitive (UPS) polymer $PEG_{5k}\text{-}b\text{-}P(DPA_{40}\text{-}r\text{-}EPA_{40}\text{-}r\text{-}AMA_3)$ and pH-insensitive polymer $PEG_{5k}\text{-}b\text{-}PEH_{80}\text{-}r\text{-}AMA_3$ were synthesized by atom transfer radical polymerization (ATRP) method reported previously. To introduce the dyes (including Cy5, Cy7.5, Ce6), the synthetic method follows the representative process described below. For Cy5 labeling, polymer (20 mg) was first dissolved in 200 μL DMF. Then Cy5-NHS ester (0.6 mg) was added and the mixture was stirred at room temperature in dark for 24 h. Then, the Cy5-conjugated polymer (UPS-Cy5) was purified by ultra-filtration to remove free dyes. The purified fluorescent polymer was freeze-dried and stored at −20 °C for further experiments. Similar protocols were utilized for the syntheses of Ce6 and Cy7.5-conjugated polymers.

**Preparation and characterization of MRIN.** Monochromatic ratiometric imaging nanoparticles (MRIN) were prepared following a previously reported procedure. A series of molecularly mixed micelles of Cy5-conjugated polymer and blank polymer with different molar ratios from 1:0 to 1:99 were prepared. Taking the molar ratio of 1:19 as an example, Cy5-conjugated polymer (1 mg) and blank polymer (19 mg) were dissolved in 1 mL methanol and then dropped into 4 mL distilled water under sonication. Methanol was removed by ultrafiltration (MWCO = 100 kDa) for 3 times. Then, the final polymer concentration was adjusted to 5 mg mL⁻¹ as a stock solution for further fluorescence characterization.

After fluorescence characterization, the molecularly mixed micelles with Cy5 fluorescence ON/OFF ratio of ~1.0 (molar ratio of Cy5-conjugated polymer to blank polymer was 1:19) were selected as the optimized formula to prepare the Always-ON micelle. The MRIN was prepared by replacing 45% of the blank polymer with Cy7.5-conjugated polymer in the above Always-ON micelle. The morphologies of micelles were visualized by transmission electron microscopy (JEM 1200EX). The particle sizes and zeta potentials of micelles were measured by Malvern Zetasizer (Nano ZSP, Malvern, UK).

**pH and light sensitivity of MRIN.** A fluorescence spectrophotometer (F-7000, Hitachi Co, Japan) was used to measure the fluorescence emission spectra of MRIN. The micelles were excited at 630 nm and 780 nm for Cy5 and Cy7.5, respectively. The corresponding fluorescence emission spectra were collected from 645 to 750 nm for Cy5, and from 795 to 850 nm for Cy7.5.

For pH sensitivity evaluation, the micelle stock solution was diluted in PBS buffer with different pH (interval 0.2 pH) to Cy5 concentration of 0.25 μg mL⁻¹. Then the fluorescence spectra at each pH were collected. For NIR laser-induced Cy7.5 photobleaching analysis, the fluorescence emission spectra of Cy5 and Cy7.5 were measured before and after 808 nm irradiation (0.5 W cm⁻², 5 min). The emission intensity at 670 nm was used to quantify the Cy5 fluorescence signal amplification, as well as the pH and light sensitivity of MRIN. For stability study, the micelle stock solution was diluted to 0.25 μg mL⁻¹ (base on Cy5) with pH 7.4 PBS, pH 5.4 PBS buffers, and plasma, respectively, and then the mixture was incubated at 37 °C and the fluorescence spectra were measured at designated time points. The fluorescence images of micelle samples were acquired using IVIS imaging system (Living Image 4.3.1, PerkinElmer, U.S.A.).

**MRIN for parsing nanoparticle microdistribution.** Three micelles with different intracellular and extracellular fluorescence properties were synthesized. In addition to the aforementioned Always-ON micelle ($RNP_{100\%}$) and MRIN, an Always-OFF ($RNP_{0\%}$) was also synthesized by self-assembly of $PEG_{5k}\text{-}b\text{-}PEH_{80}\text{-}r\text{-}Cy5$ (5%), $PEG_{5k}\text{-}b\text{-}PEH_{80}\text{-}r\text{-}Cy7.5$ (45%), and $PEG_{5k}\text{-}b\text{-}PEH_{80}\text{-}r\text{-}AMA_3$ (50%). The Cy5 signal of the MRIN was OFF at pH > pH$_t$, while turned ON at pH < pH$_t$ (pH$_t$ refers to pH transition point). However, the Cy5 signals of the $RNP_{0\%}$ and $RNP_{100\%}$ were OFF and ON during the in vivo journey, respectively.

A confocal laser scanning microscope (CLSM, NIS-Elements AR 4.20.00, A1R-Storm, Nikon, Japan) was used to visualize the intracellular and extracellular Cy5 activation of the three micelles in vitro. 4T1 cells were plated in glass-bottom dishes in 1 mL RPMI 1640 complete medium. After incubation overnight, 4T1 cells were incubated with three micelles at an equivalent concentration of Cy5 (0.25 μg mL⁻¹) for 1 h, then Hoechst 33342 and AF488-WGA were added for cell nuclei and cell membrane labeling, respectively. After 15 min incubation, the images were captured by CLSM, then the cells were irradiated in situ with an 808 nm laser at 0.5 W cm⁻² for 5 min, and the images were collected again after irradiation. In addition, the intracellular fluorescence behavior of three nanoparticles were further studied by flow cytometry (FCM, Beckman, USA). 4T1 cells were incubated with three micelles at an equivalent concentration of Cy5 (0.25 μg mL⁻¹) for 1 h, followed by removal of extracellular micelles and washing with PBS buffer for three

times. The intracellular fluorescence intensity of three nanoparticles before and after 808 nm irradiation were measured by CytoFLEX LX flow cytometer (CytExpert 2.3, Beckman, USA).

Using $RNP_{0\%}$ and $RNP_{100\%}$ as reference standards of 0% and 100% endocytosis, respectively, the quantification of MRIN endocytosis was investigated in vivo. Bilateral 4T1 tumor-bearing mice (Female BALB/c mice) were randomly divided into three groups (n = 4) and administrated intravenously with three micelles (equivalent Cy5 concentration of 75 μg kg⁻¹), respectively. At 24 h post-injection, the fluorescence images of anesthetized mice were obtained via an in vivo imaging system. Subsequently, the tumors on the right flank were irradiated in situ with an 808 nm laser for 10 min. The fluorescence images of mice were collected again after irradiation. Finally, the mice were sacrificed and tumors on the left flank (not irradiated) were immediately collected and sectioned. The fluorescence images of whole tumor slides were collected by quantitative slide scanner (Phenochart 1.0.8, PerkinElmer Vectra Polaris). Then the tumor slides were irradiated in situ with an 808 nm laser for 10 min followed by image capture again. All the ratiometric images were analyzed by Image-J 1.47 software.

**In vivo irradiation condition selection and safety.** Bilateral 4T1 tumor-bearing mice (Female BALB/c mice) were injected intravenously with MRIN (75 μg kg⁻¹ Cy5). At 24 h post-injection, the tumors on the right flank were irradiated with an 808 nm laser for 0, 2, 4, 6, 8, 10, 12 min, respectively. During irradiation, the real-time temperature of the tumor was monitored by an infrared thermal camera (Fotric 226, USA) to keep it at about 42 °C by changing the laser power, then the fluorescence images were captured immediately after each irradiation using IVIS imaging system (PerkinElmer, U.S.A.). Finally, 10 min was selected as the optimal irradiation time.

In addition, the safety of 10 min 808 nm laser irradiation was evaluated. 4T1 tumor-bearing mice were randomly divided into 3 groups, one group of mice were i.v. injected with PBS, while the other two groups of mice were i.v. injected with MRIN (Cy5 concentration of 75 μg kg⁻¹). At 24 h post-injection, the two groups of MRIN injected mice were irradiated with an 808 nm laser for 10 min, keeping the tumor temperature at about 42 °C and 52 °C, respectively. At 24 h post-irradiation, tumor tissues were excised, and cut into 10 μm sections. The tumor slides were fixed and stained with anti-CD31 antibodies (1:500). Hematoxylin and eosin (H&E) staining and Transferase-mediated dUTP nick end labeling (TUNEL) assay were performed according to the manufacturer's instructions.

**The accuracy of MRIN for parsing nanoparticle microdistribution.** Cy5-conjugated micelle, Cy7.5-conjugated micelle, and micelle cocktail (mixture of Cy5-conjugated micelle and Cy7.5-conjugated micelle) were prepared. The absorption spectra of Cy5-conjugated micelle and Cy7.5-conjugated micelle before and after 808 nm irradiation (0.5 W cm⁻², 5 min) were determined using UV − Vis spectrometer (UH5300, Hitachi Co, Japan).

The fluorescence images of Cy7.5-conjugated micelle and MRIN dispersed in pH 7.4 PBS buffer with or without irradiation were captured using IVIS imaging system (Cy5: λ$_{ex/em}$ = 640/680 nm; Cy7.5: λ$_{ex/em}$ = 745/820 nm). In addition, the bilateral 4T1 tumor-bearing mice were divided to three groups (Cy7.5-conjugated micelle, MRIN, and micelle cocktail). The right tumors were irradiated for 10 min at 24 h post-injection of micelles. The fluorescence images were collected using IVIS imaging system before and after irradiation.

**Parsing extracellular and intracellular distribution of nanoparticles in different tumors.** The female mice bearing different tumors (4T1, MCF-7, CT26, Panc02 and BxPC-3) on bilateral flanks were injected intravenously with MRIN at an equivalent Cy5 concentration of 75 μg kg⁻¹. Then fluorescence images of each mouse were collected at designated time points post-injection. After images capture, the tumors on the right flank were irradiated with an 808 nm laser for 10 min, and the fluorescence images of mice were collected again. Therefore, we can obtain the intracellular and total (intracellular + extracellular) fluorescence signals in tumor sites, enabling the calculation of the extracellular fluorescence signals by subtraction, and further quantification of the extracellular and intracellular distribution of NPs.

**pH- and light-mediated Ce6/SOG activation of MRITN.** Monochromatic ratiometric imaging/therapeutic nanoparticles (MRITN) were prepared through self-assembly of Ce6-conjugated polymer (50%) and Cy7.5-conjugated polymer (50%). To verify the FRET effect between Ce6 and Cy7.5, a fluorescence spectrophotometer (F-7000, Hitachi Co, Japan) was used to measure the emission spectrum of Ce6 (Ex: 630 nm, Em: 645-900 nm) and the excitation spectrum of Cy7.5 (Em: 825 nm). Then, the emission spectra of MRITN (UPS-Ce6/UPS-Cy7.5 hybrid micelle) and UPS-Cy7.5 micelles were also measured at 630 nm for Ce6 excitation. The pH- and light-mediated Ce6 fluorescence activation of MRITN was investigated using the similar methods as MRIN. The single oxygen generation (SOG) was estimated using N, N-Dimethyl-p-nitrosoaniline (RNO) method. In detail, MRITN (Ce6 2.5 μg mL⁻¹) was mixed with RNO and imidazole working solution, followed by 660 nm irradiation at 100 mW cm⁻² for 1 min, then the UV–Vis absorption at 440 nm (λ$_{max}$ of RNO) was recorded for quantifying single oxygen generation. The singlet oxygen generated by UPS-Cy7.5 with 808 nm irradiation (0.5 W cm⁻²,

5 min) was also measured using RNO method. The singlet oxygen generation was also determined using singlet oxygen sensor green (SOSG) reagent as the fluorescent SOG indicator.

Confocal microscopy imaging was exploited to verify the spatial activation of Ce6/SOG for MRITN at cellular level. 4T1 cells were plated in glass-bottom dishes in 1 mL RPMI 1640 complete medium and incubated overnight. For intracellular Ce6/SOG activation, 4T1 cells were incubated with MRITN (Ce6 $2.5\,\mu g\,mL^{-1}$) at 37 °C for 0.5 h, then cells were washed with PBS to remove extracellular MRITN; For extracellular Ce6/ SOG activation, the cells were incubated with MRITN at 4 °C for 0.5 h to block the cellular uptake of MRITN; For both intracellular and extracellular Ce6/SOG activation, 4T1 cells were incubated with MRITN at 37 °C for 0.5 h without removal of extracellular MRITN. Then, the cell nuclei were labeled with Hoechst 33342. The Ce6 ($\lambda_{ex}$: 633 nm) and SOSG ($\lambda_{ex}$: 488 nm) fluorescence images were captured by a Nikon confocal microscope. The cells were then irradiated in situ with an 808 nm laser at $0.5\,W\,cm^{-2}$ for 5 min, and the images were collected again. Finally, the cells were irradiated in situ with a 660 nm laser at $100\,mW\,cm^{-2}$ for 1 min, and the images were acquired in the end. The above SOSG nanoprobe for organelle SOG imaging was synthesized by our laboratory[55].

**The cell membrane binding ability of MRITN at different pH values.** 4T1 cells were incubated with MRITN (pre-irradiated with 808 nm laser) in pH 6.8 or pH 7.4 RPMI 1640 medium at 4 °C for 30 min. Then, Hoechst 33342 and AF488-WGA were added for cell nuclei and cell membrane labeling, respectively. The fluorescence images were captured by confocal microscope after washing the cells with PBS buffer twice. For flow cytometry analysis, 4T1 cells were incubated with MRITN (pre-irradiated with 808 nm laser) in pH 6.8, pH 7.1 or pH 7.4 RPMI 1640 medium at 4 °C for 30 min. Then, treated cells were washed with PBS buffer for three times, resuspended in PBS buffer and analyzed by flow cytometry.

**In vitro cytotoxicity of MRITN.** 4T1 cells were seeded in 24-well plates and treated with the following conditions, respectively. (1) Intracellular PDT group: Cells were incubated with MRITN (Ce6 concentration of $2.5\,\mu g\,mL^{-1}$, dispersed in pH 6.9 RPMI 1640 medium) at 37 °C for 0.5 h, then cells were washed with PBS to remove extracellular micelles; (2) Extracellular PDT group: Cells were incubated with MRITN (pre-irradiated with 808 nm laser) at 4 °C for 0.5 h; (3) Intracellular and extracellular PDT group: Cells were incubated with MRITN (pre-irradiated with 808 nm laser) at 37 °C for 0.5 h without removal of extracellular micelles; After the above treatments, cells were irradiated with a 660 nm laser ($100\,mW\,cm^{-2}$ for 3 min) in 4 °C under dark conditions. After 24 h incubation, the cell viability was assessed using MTT assay. Briefly, 500 μL of MTT solution ($0.5\,mg\,mL^{-1}$) were added to each well and incubated at 37 °C for 4 h. Then, the generated formazan was dissolved using 200 μL DMSO. Finally, OD values at 540 nm were measured by a microplate reader (Multiskan FC, ThermoFisher Scientific). The $IC_{50}$ values were calculated by Origin 2018 software. Additionally, the PDT induced cell apoptosis of each group was evaluated using Annexin V-FITC/PI Apoptosis Detection Kit according to the manufacturer's instructions, and analyzed by flow cytometry.

**In vivo pharmacokinetic profiles.** Five Female BALB/c mice were administrated with MRITN ($0.75\,mg\,kg^{-1}$ Ce6) via tail vein injection. Blood samples (60 μL) were collected from each mouse at 2 min, 15 min, 30 min, 1 h, 3 h, 6 h, 12 h and 24 h post-injection. Then the blood samples were centrifuged at 644 g for 10 min to obtain the plasma. Subsequently, the plasma (20 μL) was mixed with 200 μL acidified methanol and centrifuged at 9600 g for 10 min. The Ce6 fluorescence signal of the supernatant was measured by a fluorescence spectrophotometer ($\lambda_{ex/em}$ = 400 nm/670 nm).

**In vivo antitumour efficacy of maximized PDT.** Female BALB/c mice bearing 4T1 tumors were randomly divided into 6 groups ($n = 9$), one group of mice were i.v. injected with PBS, while the other five groups of mice were i.v. injected with MRITN at an equivalent Ce6 concentration of $0.75\,mg\,kg^{-1}$ on day 1 and day 5. At 3 h post-injection, the five groups of MRITN injected mice were treated with no irradiation, 808 nm irradiation, 660 nm irradiation, 660 nm irradiation followed by 808 nm irradiation, and 808 nm irradiation followed by 660 nm irradiation, respectively. Laser irradiation at 808 nm was performed for 10 min, and the tumor temperature was maintained at about 42 °C during irradiation. Laser irradiation at 660 nm was performed at $400\,mW\,cm^{-2}$ for 10 min. Tumor volumes and weights were measured every other day since initial treatment. The survival curve of 4T1-bearing mice was recorded during the antitumour study. At 24 h after the last treatment, one mouse from each group was euthanized and the tumors were excised for histological analysis. Animals were sacrificed when the tumor volumes reached 1500 mm³ (height × width² × 0.5). In a separate experiment, 4T1-bearing mice were randomly divided into 6 groups ($n = 7$) and treated as aforementioned. At the end of the antitumour study, the mice were euthanized, and the tumors were collected and photographed. The major organs were also excised for histological analysis. The antitumour study of MRITN with drug-light interval of 24 h was also performed. Mice were irradiated with 660 nm laser at $400\,mW\,cm^{-2}$ for 10 min at 24 h post-injection of MRITN (Ce6 dose of $0.75\,mg\,kg^{-1}$). The PDT treatment was given three times separately at day 1, day 5 and day 9 ($n = 6$).

**Histological analysis and biosafety evaluation.** The excised tumors in above antitumour study section were cut into 10 μm slices, fixed in 4% paraformaldehyde, permeabilized with 0.1% Triton X-100, and blocked with quick blocking solution. Subsequently, the slices were stained with Rat anti-ER-TR7 antibody (1:40) at 4 °C overnight. Then AF594-conjugated Goat anti-Rat secondary antibody (1:1000) was added and incubated at 37 °C for 2 h. Finally, nuclei were stained with Hoechst 33342. The tumor slices were also stained with collagen and fibronectin antibodies, respectively. H&E staining and TUNEL assay were performed as well. The whole mount images of tumor slices were collected by a quantitative slide scanner.

To evaluate the biosafety of MRITN, the major organs (*e.g.*, heart, liver, spleen, lung, and kidney) collected in above antitumour study section were fixed in 4% paraformaldehyde, mounted with paraffin, sliced into 10 μm, and then examined by H&E staining.

**Antitumour metastasis study of maximized PDT.** BALB/c mice were injected with $1 \times 10^6$ luciferase-transfected 4T1 cells on the right flank. Six days later, the mice were randomly divided into 3 groups ($n = 5$) and injected intravenously with PBS or MRITN (Ce6 $0.75\,mg\,kg^{-1}$). At 3 h post-injection, the two groups of MRITN injected mice were treated with either 660 nm irradiation or 808 nm irradiation followed by 660 nm irradiation. The tumor developments of the mice were monitored by bioluminescence imaging (PerkinElmer, U.S.A.). The lung metastasis nodules were examined using bioluminescence imaging at 22 days after treatment. Finally, the mice were sacrificed and lung tissues were collected for ex vivo bioluminescence imaging and H&E staining.

**Analysis of integrin β1 expression.** 4T1 and MCF-7 cells were treated with the following conditions: (1) PBS group; (2) Intracellular PDT group: Cells were incubated with MRITN at 37 °C for 0.5 h, then cells were washed with PBS to remove extracellular micelles; (3) Extracellular PDT group: Cells were incubated with MRITN (pre-irradiated with 808 nm laser) at 4 °C for 0.5 h. Then the cells were irradiated at 660 nm at $50\,mW\,cm^{-2}$ for 2 min (mild PDT with >85% cell survival). Subsequently, the integrin β1 expression of different groups was analyzed by FCM, CLSM, and western blot (WB), respectively. In detail, the treated cells were fixed, stained with Mouse anti-integrin β1 antibodies (ab24693, 1:200), and then incubated with AF594-conjugated Goat anti-Mouse (ab96873, 1:1000) secondary antibody, measured by FCM and CLSM, separately. For WB analysis, the treated cells were washed with PBS and then lysed with RIPA lysis buffer (150 mM NaCl, 0.5% DOC, 25 mM pH 7.4 Tris, 0.1% SDS, 0.1% Triton-X100) supplemented with a phosphatase/protease inhibitor cocktail. After electrophoresis, the proteins were transferred to 0.45-μm PVDF membrane, blocked for 2 h with 5% nonfat milk, and incubated with primary antibodies against Rabbit anti integrin β1 (ab179471,1:2000) and Mouse anti-α-tubulin (T5168, 1:10,000) overnight. Then the transferred membranes were washed for four times with TBST, conjugated with HRP-labeled Goat anti-Rabbit (ab6721, 1:5000) and Goat anti-Mouse (ab6789, 1:10000) secondary antibodies at room temperature for 2 h, and finally detected by ECL chemiluminescence.

**Statistics and reproducibility.** The photographic and TEM images of MRIN in Fig. 2b and Supplementary Fig. 2e were repeated thrice independently with similar results, and one representative image from each group was shown. The pH transition curve of Fig. 2c was measured thrice independently with similar results. Confocal imaging of 4T1 cells incubated with $RNP_{0\%}$, MRIN, $RNP_{100\%}$, and MRITN at pre- and post-irradiation were repeated at least three times with similar results, and a series of representative images from each group were shown, such as Figs. 3b, 5a, and Supplementary Figs. 4a, 14a. In vivo imaging of mice at post-injection of $RNP_{0\%}$, MRIN, $RNP_{100\%}$, and MRITN was repeated at least three times using biologically independent mice with similar results, and a series of representative images were shown, such as Figs. 3c and 4a, e, and Supplementary Figs. 9, 10, 15a. For the intracellular, extracellular, and total distribution of nanoparticles in tumor sections, as well as the antibody stain, H&E staining and TUNEL staining of tumor slices, the experiment was repeated thrice with similar results, a series of representative images were shown, such as Figs. 3e, 4d and 6d, and Supplementary Figs. 5f, 7, 17a, 17c, 18.

**Statistical analysis.** Data were shown as means ± s.d. Statistical significance was assessed using unpaired Student's *t*-test or two-way analysis of variance (ANOVA) with Tukey's post-hoc test according to the number of groups and variables. Survival curves were compared by a log-rank test. All statistical analyses were performed in GraphPad Prism 7.0 software.

**Reporting summary.** Further information on research design is available in the Nature Research Reporting Summary linked to this article.

## Data availability
The authors declare that the data supporting the findings of this study are available within the article, source data, and its Supplementary Information. The source data underlying Figs. 2, 3, 4, 5, 6, Supplementary Figs. 2, 4, 6, 13, 14, 15, 16, 17, and western bot are provided with this paper. A reporting summary for this article is available as a Supplementary Information file. Source data are provided with this paper.

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

## Acknowledgements

This research was financially supported by the National Key Research and Development Program of China (2017YFA0205600 to Y.W.), the National Natural Science Foundation of China (NSFC) grants (81973260 to Y.W. and 81903554 to B.C.), and the Beijing Natural Science Foundation (JQ19025 to Y.W.). We thank the State Key Laboratory of Natural and Biomimetic Drugs, Peking University Biological Imaging Facility for confocal, animal, and tissue imaging services.

## Author contributions

Y.Yan and Y.G.W. are responsible for all the phases of the project. B.C. optimized the formulations and interpreted the results; Q.Y. helped to perform the in vivo imaging; Z.W. synthesized the polymer; Y.Yang and F.W. assisted on animal work and western blot; Y.Q.W. and M.T. helped with the cell culture; H.X., M.C., and J.L. assisted with fluorescence measurement; S.W. and Q.Z. provided useful suggestions; Y.Yan wrote the paper; Y.G.W. and B.C. revised the manuscript.

## Competing interests

The authors declare no competing interests.
