## [Peer Review File · Nature Communications]

Dissecting extracellular and intracellular distribution of nanoparticles and their contribution to therapeutic response by monochromatic ratiometric imagingREVIEWER COMMENTS

Reviewer #1 (expertise: PDT and tumour microenvironment)- Remarks to the Author:

Comments to the Authors:

The authors submit an interesting and important study that leverages a nanoconstruct based formulation for quantifying tumor distribution of nanoparticles and exploiting it for photodynamically targeting both intracellular and extracellular targets for a better treatment outcome. Although the study is an interesting one, there are however several concerns that have to be addressed.

- The authors throughout the manuscript mention “extracellular nanoparticles” for the ones that are activated through an external trigger i.e NIR irradiation at 808 nm. However, it appears that a fraction of nanoparticles is internalized, yet not activated in the lysosomes. The authors should provide a discussion on that as well. For example, in figure 3b all the three nanoparticle formulations RNP0%, MRIN and RNP100% show a diffused intracellular fluorescence, yet the authors mention it as intracellular and extracellular fluorescence. It will be appropriate to quantify intracellular fluorescence through flow cytometry with and without irradiation of a cell suspension treated with the RNP0%, MRIN and RNP100%.
- In figure 3d the quantification of percentage extracellular and intracellular nanoparticle fluorescence is confusing. It appears to be the reverse of what is depicted in figure 3c.
- For quantification of intra and extracellular of nanoparticles, the authors use a 24 h time-point. However, for PDT studies the authors prefer a 3 h drug-light interval which is surprising given the relatively lower intracellular nanoparticle content at this time point (Supplementary figure 10a and 10b).
- For the in vivo PDT treatment study, the authors use a PDT scheme with two PDT treatments on day 1 and day 5. Why was this treatment strategy followed and what were the tumor characteristics (size and volume) on day 1.
- The authors suggest that the reduction in ECM and integrin expression of tumor cells could be possible reasons for reduction in metastasis. While this may be the case, the authors should discuss the negative consequences of ECM degradation which has been previously reported to enhance tumor metastasis. (1). Stromal Elements Act to Restrain, Rather Than Support, Pancreatic Ductal Adenocarcinoma, *Cancer Cell*, Volume 25, Issue 6, 16 June 2014, Pages 735-747. (2). Depletion of Carcinoma-Associated Fibroblasts and Fibrosis Induces Immunosuppression and Accelerates Pancreas Cancer with Reduced Survival, *Cancer Cell*, Volume 25, Issue 6, 16 June 2014, Pages 719-734.
- The authors refer to “mild extracellular PDT” as a cause of ECM disruption and integrin down regulation. The authors should include discussion on photodynamic priming which in general includes the effects of therapeutic and sub-therapeutic doses of PDT. (1). Impacting Pancreatic Cancer Therapy in Heterotypic in Vitro Organoids and in Vivo Tumors with Specificity-Tuned, NIR-Activable Photoimmunonanoconjugates: Towards Conquering Desmoplasia?, *Nano Lett.* 2019, 19, 11, 7573–7587. (2). Subtherapeutic Photodynamic Treatment Facilitates Tumor Nanomedicine Delivery and Overcomes Desmoplasia, *Nano Lett.* 2021, 21, 1, 344–352. (3). Photodynamic therapy, priming and optical imaging: Potential co-conspirators in treatment design and optimization, *Journal of Porphyrins and Phthalocyanines*. Vol. 24, No. 11n12, pp. 1320-1360 (2020).
- The manuscript needs editing for grammar. For example, “The Cy5 signals of tumors greatly increased along with the disappeared Cy7.5 fluorescence at each time point after irradiation.”, “The increased Cy5 signals at the tumor sites before and after irradiation were all originated from extracellular nanoparticles”, “The tumor slices were also performed for collagen, fibronectin, and H&E staining as well as TUNEL assay, respectively”, etc.

Reviewer #2 (expertise: Ratiometric imaging)- Remarks to the Author:

In this manuscript entitled “Dissecting extracellular and intracellular distribution of nanoparticles and their contribution to therapeutic response by monochromatic ratiometric imaging”, Prof. Wang and collaborators developed a pH/light dual-responsive monochromatic ratiometric imaging nanoprobe (MRIN) for accurately quantifying the extracellular and intracellular distribution of NPs

in several tumor models. This study indeed offers a valuable tool to visualize and dissect the contribution of extracellularly distributed nanophotosensitizer to therapeutic efficacy and maximize the treatment outcome of PDT. Overall, this study has a high impact on drug delivery and tumor theranostics, the experiments were well designed and carefully conducted with proper controls and the hypothesis and claims were scientifically vigorous. Thus, I recommend its publication after addressing the following issues.

1. The stability of micelles depends on the CMC of copolymers. The authors should make sure that the MRIN keep self-assembled nanostructure in vivo before internalization.
2. To quantify the extracellular and intracellular distribution of MRIN in tumor, the imaging of mice was conducted for 48 h, which showed the highest fluorescence signals of intracellular and extracellular nanoprobe. A longer time monitoring should be conducted to investigate the profile of microdistribution in tumors.
3. The cyanine dyes can also work as photosensitizers (Pharmaceutics 2021, 13, 818). A mild elevated SOSG signal was observed after 808 nm irradiation in Fig. 5a. The authors should rule out the effect of Cy7.5 on photodynamic therapy.
4. The quantitative data of lung metastasis in Fig.6b should be provided for significant difference analysis.
5. The photograph and TEM image of MRIN at pH 5.4 with 808 nm irradiation should be provided in Figure 2b for direct comparison.
6. A scheme should be provided to describe the "turn-on" mechanism of MRIN nanoprobe regarding pH-induced dissociation and 808 nm laser irradiation, respectively.

Reviewer #3 (expertise: pH sensitive fluorescent nanoparticles)- Remarks to the Author:

This manuscript described an approach on dissecting extracellular and intracellular distribution of nanoparticles. The authors developed a pH/light dual-response monochromatic ratiometric-imaging nanoparticle (MRIN), which precisely quantified nanocarrier micro-distribution. Particularly, the MRIN exhibited a sharp pH response and high Cy5 signal activation ratio. The application demonstrated in the manuscript has relevance and potential high impact in the field of imaging and photodynamic therapy of tumors. Several revisions are suggested aiming to improve quality of the results and presentation.

1. According to the experimental procedure, the MRIN was formed by self-assembly of components with a fixed proportion, how to control such uniform size as shown in Figure 2b, and the author should provide a TEM of with high resolution.
2. The author chose RNP0%, RNP100% as positive controls. According to the design, RNP100% as an always-on probe has no quenching effect of Cy7.5 without pH responsive, why there are good targeted imaging results in Figure 3c.
3. Does the principle total=intracellular + extracellular applicable to MRIN also apply to RNP0%, RNP100%? For RNP100%, There is no acid response-mediated fluorescence recovery, so it is not certain that exhibited fluorescence means intracellular, extracellular nanoparticles also exhibited fluorescence. Similarly, the RNP0% did not produce fluorescence, which does not mean that they were extracellular, the author needs to provide a reasonable explanation or correct the description in Figure 3b.
4. Four kinds of particles, MRIN, RNP0%, RNP100%, and MRITN are used in the manuscript, and the self-assembly components include UPS, UPS-Cy5, UPS-Cy7.5, pH-insensitive polymer, pH-insensitive polymer-Cy5, pH-insensitive polymer-Cy7.5, UPS-Ce6. Although components and ratio were listed separately in the experimental steps, it is not very clearly stated in the text. The self-assembly composition and ratio of each particles may be listed in a table or in the corresponding figure.
5. Why Cy7.5 also quenched Ce6? the author needs to provide FRET related explanation.
6. Regarding why PDT with intracellular and extracellular is better, the author needs to provide a mechanism explanation or further pathway data to support Figure 5.

Point-by-point response to reviewers

We would like to thank the reviewers for the insightful and constructive comments! We have revised the manuscript according to their advices, which should significantly improve the clarity and quality of our work. Below is a list of the point-by-point responses to the reviewer comments shown in *italics* and the corresponding changes that we made highlighted in **yellow**.

Reviewer #1 (Remarks to the Author):

The authors submit an interesting and important study that leverages a nanoconstruct based formulation for quantifying tumor distribution of nanoparticles and exploiting it for photodynamically targeting both intracellular and extracellular targets for a better treatment outcome. Although the study is an interesting one, there are however several concerns that have to be addressed.

We appreciate the reviewer's encouraging comments on the novelty and impact of the MRIN nanotechnology. We have revised the manuscript according to your advices and other referee's comments.

1. *The authors throughout the manuscript mention "extracellular nanoparticles" for the ones that are activated through an external trigger i.e NIR irradiation at 808 nm. However, it appears that a fraction of nanoparticles is internalized, yet not activated in the lysosomes. The authors should provide a discussion on that as well. For example, in figure 3b all the three nanoparticle formulations RNP_{0%}, MRIN and RNP_{100%} show a diffused intracellular fluorescence, yet the authors mention it as intracellular and extracellular fluorescence. It will be appropriate to quantify intracellular fluorescence through flow cytometry with and without irradiation of a cell suspension treated with the RNP_{0%}, MRIN and RNP_{100%}.*

Re: Thanks for the reviewer's good suggestion. We have performed new experiments to quantify intracellular fluorescence by flow cytometry and added the result as **Supplementary Figure 4**. Luminal pH decline from neutral to mild acidic (e.g. pH 6.0 for early endosome, pH 5.0 for lysosome) is a hallmark of cellular internalization. In our previous study, we have succeeded to target specific endocytic organelles with different pH values by precisely tuning the pH transitions (pH_t) of the UPS fluorescent nanoprobe library (Wang *et al*, *Adv Mater* **2017**, 29, 1603794). In this manuscript, we developed a pH/light dual-responsive monochromatic ratiometric imaging nanotechnology (MRIN) to dissect extracellular and intracellular distribution of nanoparticles in tumor tissues. The pH transition of MRIN is 6.3, which can be activated in early endosomes (pH~6.0) upon internalization (Wang *et al*, *Nat Mater*, **2014**, 13, 204; Zhou *et al*, *Angew Chem Int Ed Engl* **2011**, 50, 6109 -6114). Due to the ultra-pH-sensitivity of MRIN, it can quickly dissociate into unimers with exponential Cy5 signal amplification into early endosome after being endocytosed (~5 min). As shown in **Supplementary Figure 4**, negligible enhancement of intracellular Cy5 signal was observed after 808 nm laser irradiation, demonstrating that the internalized MRIN was almost completely activated in the endo-lysosomes.

In order to demonstrate the quantitative imaging feasibility of MRIN, RNP_{0%} (Cy5-labeled pH-insensitive micelle) and RNP_{100%} (Always-ON micelle) were established to simulate the artificial states of 0% and 100% endocytosis *in vitro* and *in vivo*. However, it doesn't mean that they are true 0% and 100% endocytosis. The Cy5 signal of RNP_{0%} was completely 'OFF' whether cellular endocytosis or not due to its pH-insensitive nanostructure, so it could be used to simulate the artificial states of 0% endocytosis. The Cy5 signal of RNP_{100%} was completely 'ON' due to the Always-ON design, so it could be used to simulate the artificial states of 100% endocytosis. The intracellular Cy5 signal of RNP_{0%} was completely activated after 808 nm irradiation with over 40-fold signal amplification. In contrast, the intracellular Cy5 signal of RNP_{100%} kept constant after 808 nm laser irradiation due to the Always-ON design.

Line 104-106: For MRIN, the Cy5 signal of which endocytosed into the cells was fully activated (Supplementary Fig. 4a),

Line 121-126: Furthermore, the intracellular fluorescence behavior of three nanoparticles was also studied by flow cytometry (Supplementary Fig. 4b). The intracellular Cy5 signal of RNP_{0%} was completely activated after 808 nm irradiation with over 40-fold signal amplification. Whereas, the intracellular Cy5 signal of RNP_{100%} kept constant after 808 nm laser irradiation due to the Always-ON design. For MRIN, negligible enhancement of intracellular Cy5 signal was observed after 808 nm laser irradiation, demonstrating that the internalized MRIN were completely activated in the endo-lysosomes.

Supplementary Figure 4. The intracellular fluorescence behavior of three nanoparticles. (a) The intracellular Cy5 fluorescence images of MRIN in 4T1 cells treated with MRIN before and after 808 nm irradiation. **(b)** The intracellular fluorescence intensity of three nanoparticles in 4T1 cell suspension before and after 808 nm irradiation measured by flow cytometry.

2. In figure 3d the quantification of percentage extracellular and intracellular nanoparticle fluorescence is confusing. It appears to be the reverse of what is depicted in figure 3c.

Re: We appreciate the reviewer for pointing it out. We are very sorry for our mistake and have made correction in Figure 3d according to your comments.

3. For quantification of intra and extracellular of nanoparticles, the authors use a 24 h time-point. However, for PDT studies the authors prefer a 3 h drug-light interval which is surprising given the relatively lower intracellular nanoparticle content at this time point (Supplementary figure 10a and 10b).

Re: We appreciate the insightful questions by the reviewer. As the reviewer mentioned, the intracellular nanoparticle content at 3 h post-injection is relative lower than that at 24 h. However, we chose the drug-light interval (DLI) of 3 h to perform the anti-tumor PDT studies, because that we have discovered that the anti-tumor efficacy of intracellular PDT at DLI of 3 h is much better than that of 24 h. The detailed studies and corresponding mechanisms have been submitted and revised on *Nature Nanotechnology*.

In addition, we added the anti-tumor results of MRITN at DLI of 24 h that performed before the first submission of the manuscript. The result was in consistence with that of DLI of 3 h. The anti-tumor efficacy of MRITN with or without 808 nm laser irradiation groups has no significant difference as compared with PBS group. The antitumor efficacy of MRITN combined with 660 nm laser group (intracellular PDT) and MRITN combined with 660 nm + 808 nm group was significantly enhanced as compared with MRITN plus 808 nm group. However, harnessing 808 nm irradiation before 660 nm (intracellular + extracellular PDT) resulted in the most efficient antitumor efficacy. We have added the related results as **Supplementary Figure 16**.

Line 256-258: In addition, the antitumour study of MRITN with DLI of 24 h was also performed.

Similarly, combined PDT achieved the most effective tumour inhibition as compared with other treatment groups (Supplementary Fig. 16).

Supplementary Figure 16. The anti-tumor study of MRITN with drug-light interval of 24 h.

4. For the *in vivo* PDT treatment study, the authors use a PDT scheme with two PDT treatments on day 1 and day 5. Why was this treatment strategy followed and what were the tumor characteristics (size and volume) on day 1.

Re: The reviewer raises a good point. Firstly, we performed anti-tumor study when the tumor volumes of mice reached approximately 50-100 mm³, the first day of anti-tumor study was defined as day 1. Secondly, in this manuscript, we developed a nanoparticle (MRITN) which could realize combined intracellular and extracellular PDT, aiming to reduce PDT dose (including photosensitizer dose and laser power) and prolong administration intervals for improved compliance without compromising the therapeutic effect.

Currently, a majority of activatable PDT rely on the intracellular exposure to exert lethal tumor damage, which requires high photosensitizer dose or short administration intervals to achieve the desired therapeutic effect. Many literatures for photodynamic therapy performed treatment every 2-3 days for at least three times (Gao A, *et al*, *Nano Lett* **2020**, 20, 353-362; Su J, *et al*, *Theranostics* **2017**, 7, 523-537; Duan X, *et al*, *JACS* **2016**, 138, 16686-16695). Enabled by MRITN, we achieved efficient antitumor efficacy by two PDT treatments with 4 days intervals using 6-fold lower photosensitizer dose than most reported studies for PDT therapy (Wang T, *et al*, *ACS nano* **2016**, 10, 3496-3508; Fu Y, *et al*, *J Control Release* **2020**, 327, 129-139).

5. The authors suggest that the reduction in ECM and integrin expression of tumor cells could be possible reasons for reduction in metastasis. While this may be the case, the authors should discuss the negative consequences of ECM degradation which has been previously reported to enhance tumor metastasis. (1). *Stromal Elements Act to Restrain, Rather Than Support, Pancreatic Ductal Adenocarcinoma*, *Cancer Cell*, Volume 25, Issue 6, 16 June 2014, Pages 735-747. (2). *Depletion of Carcinoma-Associated Fibroblasts and Fibrosis Induces Immunosuppression and Accelerates Pancreas Cancer with Reduced Survival*, *Cancer Cell*, Volume 25, Issue 6, 16 June 2014, Pages 719-734.

Re: Thanks for the reviewer's good suggestion, we have discussed the negative consequences of ECM degradation in the Discussions and cited related literatures.

Line 321-327: Based on our findings, we speculate that the potential mechanism for the anti-metastasis effect of combined PDT is the ECM destruction and the downregulation of adhesion integrin $\beta 1$. However, the tumour metastasis is a very complicated process, and influenced by various physiological factors. Although previous investigations have reported that excessive reduction of extracellular matrix and stromal

cells promotes tumour metastasis^{53, 54}, our MRITN-mediated PDT achieved remarkable inhibition of lung metastasis probably due to the lethal damage to tumour cells. However, the comprehensive mechanism needs to be further investigated in the future.

6. The authors refer to “mild extracellular PDT” as a cause of ECM disruption and integrin down regulation. The authors should include discussion on photodynamic priming which in general includes the effects of therapeutic and sub-therapeutic doses of PDT. (1). *Impacting Pancreatic Cancer Therapy in Heterotypic in Vitro Organoids and in Vivo Tumors with Specificity-Tuned, NIR-Activable Photoimmunonanoconjugates: Towards Conquering Desmoplasia?*, *Nano Lett.* 2019, 19, 11, 7573–7587. (2). *Subtherapeutic Photodynamic Treatment Facilitates Tumor Nanomedicine Delivery and Overcomes Desmoplasia*, *Nano Lett.* 2021, 21, 1, 344–352. (3). *Photodynamic therapy, priming and optical imaging: Potential co-conspirators in treatment design and optimization*, *Journal of Porphyrins and Phthalocyanines*. Vol. 24, No. 11n12, pp. 1320-1360 (2020).

Re: Thanks for the reviewer’s suggestion, we have added a discussion on photodynamic priming in the Discussions and cited related literatures.

Line 309-316: Currently, photodynamic therapy (PDT) has been widely exploited for the treatment of various malignant tumours due to its minimal invasion and fast healing. The responses to PDT rely largely on the light dose, photosensitizer concentration and location⁵⁰. In addition to the vascular damage and direct cell killing effect at therapeutic dose level, photodynamic priming (PDP) using subtherapeutic dose has also been demonstrated to efficiently enhance the antitumour efficacy of subsequent therapies. Several researches have revealed that PDP can sensitize tumours to immuno- and chemo-therapy via the tumour microenvironment modulation, including the decrease of extracellular matrix content and the enhancement of tumour vascular leakiness^{51, 52}.

Line 317-320: Our results demonstrated that the extracellular NPs played an equal important role in cancer treatment via both cell membrane damage (direct cell killing) and ECM destruction (photodynamic priming), enabling the maximized therapeutic outcomes of combined PDT.

7. The manuscript needs editing for grammar. For example, “The Cy5 signals of tumors greatly increased along with the disappeared Cy7.5 fluorescence at each time point after irradiation.”, “The increased Cy5 signals at the tumor sites before and after irradiation were all originated from extracellular nanoparticles”, “The tumor slices were also performed for collagen, fibronectin, and H&E staining as well as TUNEL assay, respectively”, etc.

Re: Thank you very much for your kind advice. We have checked the grammar carefully and made some changes in the revised manuscript. These changes will not influence the content and framework of the paper. We appreciate for reviewer’s warm work earnestly, and hope that the correction will meet with approval.

Reviewer #2 (Remarks to the Author):

In this manuscript entitled “Dissecting extracellular and intracellular distribution of nanoparticles and their contribution to therapeutic response by monochromatic ratiometric imaging”, Prof. Wang and collaborators developed a pH/light dual-responsive monochromatic ratiometric imaging nanoprobe (MRIN) for accurately quantifying the extracellular and intracellular distribution of NPs in several tumor models. This study indeed offers a valuable tool to visualize and dissect the contribution of extracellularly distributed nanophotosensitizer to therapeutic efficacy and maximize the treatment outcome of PDT. Overall, this study has a high impact on drug delivery and tumor theranostics, the experiments were well designed and carefully conducted with proper controls and the hypothesis and claims were scientifically vigorous. Thus, I recommend its publication after addressing the following issues.

We thank the reviewer's positive comments. We provide a point-by-point response to the reviewer comments.

1. The stability of micelles depends on the CMC of copolymers. The authors should make sure that the MRIN keep self-assembled nanostructure in vivo before internalization.

Re: We acknowledge the reviewer's concern. Our previous article (Wang *et al*, *Nat Mater*, **2014**, 13, 204) has investigated the dilution of ultra-pH-sensitive micelle in blood after intravenous injection. The plasma concentration of micelle at 24 h after injection was approximately $100 \mu\text{g mL}^{-1}$ (calculated from the 10 mg kg^{-1} injection dose and micelle pharmacokinetics), which was exponentially higher than the CMC of copolymers (The CMC was ranging from $0.88\text{-}2.48 \mu\text{g mL}^{-1}$ for different ultra-pH-sensitive copolymers). We have also measured the CMC of our nanoparticle to be $1.97 \mu\text{g mL}^{-1}$ by a pyrene assay. According to the *in vivo* pharmacokinetics of our nanoparticle (**Supplementary Figure 15c**) and 60 mg kg^{-1} injection dose, the plasma concentration of our nanoparticle at 2 min and 24 h after intravenous injection were $600 \mu\text{g mL}^{-1}$ and $107.4 \mu\text{g mL}^{-1}$, respectively, which were dozens of times higher than the CMC. Therefore, MRIN can remain stability in the blood circulation and will not disassemble until they are internalized by tumor cells.

2. To quantify the extracellular and intracellular distribution of MRIN in tumor, the imaging of mice was conducted for 48 h, which showed the highest fluorescence signals of intracellular and extracellular nanoprobe. A longer time monitoring should be conducted to investigate the profile of microdistribution in tumors.

Re: Per the reviewer's advice, we have carried out new experiment for long-term monitoring the extracellular and intracellular distribution of MRIN *in vivo* using 4T1 tumor-bearing mice. As shown in the **new Figure 4a-c**, intracellular, extracellular, and total nanoparticle exposure in tumors all increased gradually over 48 h, which implied nanoprobe in circulation extravasated unceasingly from leaky tumor vasculature and then were internalized into intratumoral cells. In the following 2 days, the intracellular nanoparticle exposure remained unchanged, while the extracellular and total nanoparticle exposure were gradually decreased, which probably due to the slow clearance of nanoparticles from mice. Accordingly, the extracellular nanoparticle exposure percentage and extracellular/intracellular exposure ratio reached a maximum at 48 h post-injection followed by a slight decrease in the subsequent monitoring period.

Line 187-193: The quantitative results revealed that the exposure level of nanoparticles in intracellular and extracellular compartments of tumour tissues was continuously enhanced within 48 h post-injection. However, the intracellular nanoparticle exposure remained unchanged, while the extracellular nanoparticle exposure was gradually decreased in the following 2 days, which probably due to the slow clearance of nanoparticles from mice (Fig. 4b). Accordingly, the percentage of extracellular nanoparticle exposure and

the ratio of extracellular versus intracellular exposure reached a maximum at 48 h post-injection followed by a slight decrease in the subsequent monitoring period.

Fig. 4 Quantitative imaging of extracellular and intracellular distribution of nanoparticles in different tumors.

3. The cyanine dyes can also work as photosensitizers (Pharmaceutics 2021, 13, 818). A mild elevated SOSG signal was observed after 808 nm irradiation in Fig. 5a. The authors should rule out the effect of Cy7.5 on photodynamic therapy.

Re: Thanks for the reviewer’s good suggestion, we have carried out new experiments to investigate the effect of Cy7.5 on photodynamic therapy and added the related results as **Supplementary Figure 13**. For MRITN, Cy7.5 served as a fluorescence quencher of Ce6 through FRET effect. At pH 7.4, MRITN is micelle state with Ce6 signal “OFF”, therefore, it generates little singlet oxygen under 660 nm irradiation. The Ce6 signals of MRITN can be fully activated through both pH-induced micelles dissociation and 808 nm irradiation-induced Cy7.5 photobleaching, resulting in about 270-fold higher SOG than Ce6 “OFF” state under 660 nm irradiation. However, the singlet oxygen generated by UPS-Cy7.5 with 808 nm irradiation was 20-fold lower than MRITN with 660 nm irradiation in activated Ce6 state. Therefore, the SOG of Cy7.5 was negligible for the photodynamic effect of MRITN. In addition, for the photodynamic effect at cellular level, MRITN was pre-irradiated with 808 nm laser before incubation with cells, aiming at ruling out the effect of Cy7.5 on photodynamic therapy. What’s more, for *in vivo* anti-tumor study, the mice treated with MRITN plus 808 nm irradiation was also included as a control group. The results showed that 808 nm irradiation alone had no significant difference as compared with PBS group on tumor inhibition, further confirming the insignificant photodynamic effect of Cy7.5.

Line 220-223: Although Cy7.5 can also work as photosensitizers, its singlet oxygen generation under 808 nm irradiation was 20-fold lower than MRITN under 660 nm irradiation. Hence, compared with Ce6, the SOG of Cy7.5 was negligible for the photodynamic effect of MRITN (Supplementary Fig. 13).

Supplementary Figure 13. The effect of Cy7.5 on photodynamic therapy.

4. The quantitative data of lung metastasis in Fig.6b should be provided for significant difference analysis.

Re: We acknowledge the reviewer's kind suggestion. We have added the quantitative data of lung metastasis in **Figure 6c**, and performed significant difference analysis.

Figure 6c. The quantitative result of *ex vivo* lung metastasis.

5. The photograph and TEM image of MRIN at pH 5.4 with 808 nm irradiation should be provided in Figure 2b for direct comparison.

Re: Per the reviewer's advice, we have added the photograph and TEM image of MRIN at pH 5.4 with 808 nm irradiation in **Figure 2b**.

Figure 2b. The photographic images and TEM images of MRIN.

6. A scheme should be provided to describe the “turn-on” mechanism of MRIN nanoprobe regarding pH-induced dissociation and 808 nm laser irradiation, respectively.

Re: We appreciate the reviewer’s suggestion. In the MRIN platform, Cy7.5 served as a fluorescence quencher of Cy5 through FRET effect. Tertiary amino groups are incorporated into polymers as ionizable groups to impart pH sensitivity. When $\text{pH} > 6.3$, the hydrophobic segments of PEG_{5k}-*b*-P(DPA_{40-r}-EPA_{40-r}-Dye) polymer self-assemble into the micelle cores, leading to fluorescence quenching of Cy5 fluorophore by hetero-FRET. In micelle state, the Cy5 fluorescence signal can be recovered by photobleaching Cy7.5 with 808 nm irradiation. When $\text{pH} < 6.3$, protonation of the PEG_{5k}-*b*-P(DPA_{40-r}-EPA_{40-r}-Dye) segments results in micelle dissociation, leading to abolishment of FRET effect due to the long distance between fluorophore and fluorophore quencher. Therefore, both pH-induced dissociation and 808 nm laser irradiation can effectively turn on MRIN nanoprobe. We have included the scheme in **Supplementary Figure 3**.

Line 94-96: The ‘turn-on’ mechanisms of MRIN based on pH-induced dissociation and 808 nm laser irradiation are shown in Supplementary Fig. 3.

Supplementary Figure 3. The ‘turn-on’ mechanism of MRIN based on pH-induced dissociation and 808 nm laser irradiation.

Reviewer #3 (Remarks to the Author):

This manuscript described an approach on dissecting extracellular and intracellular distribution of nanoparticles. The authors developed a pH/light dual-response monochromatic ratiometric-imaging nanoparticle (MRIN), which precisely quantified nanocarrier micro-distribution. Particularly, the MRIN exhibited a sharp pH response and high Cy5 signal activation ratio. The application demonstrated in the manuscript has relevance and potential high impact in the field of imaging and photodynamic therapy of tumors. Several revisions are suggested aiming to improve quality of the results and presentation.

We appreciate the thorough and thoughtful comments by the reviewer. We provide a point-by-point response to the reviewer comments.

1. According to the experimental procedure, the MRIN was formed by self-assembly of components with a fixed proportion, how to control such uniform size as shown in Figure 2b, and the author should provide a TEM of with high resolution.

Re: Thanks for the reviewer's positive comments on our nanoparticles. As the reviewer mentioned, the MRIN was a self-assembly of three components with a fixed proportion for PEG_{5k}-*b*-P(DPA₄₀-*r*-EPA₄₀)-*r*-Cy5 (5%), PEG_{5k}-*b*-P(DPA₄₀-*r*-EPA₄₀-*r*-Cy7.5) (45%), and PEG_{5k}-*b*-P(DPA₄₀-*r*-EPA₄₀) (50%). The three components are from the same copolymer. Because of the high molecular weight (Mw) of ultra-pH-sensitive copolymer (~22.0 kDa), the conjugation of small molecules, such as Cy5 (Mw: 616) and Cy7.5 (Mw: 782) have negligible effect on the physicochemical properties of copolymer. Therefore, the MRIN can be prepared by a one-step self-assembly procedure. In addition, sonication dispersion method was applied to prepare MRIN, enabling the relatively uniform particle size. According to the reviewer's advice, we have provided the TEM of MRIN with high resolution in **Supplementary Figure 2e**.

Supplementary Figure 2e. The TEM images of MRIN with high resolution.

2. The author chose RNP_{0%}, RNP_{100%} as positive controls. According to the design, RNP_{100%} as an always-on probe has no quenching effect of Cy7.5 without pH responsive, why there are good targeted imaging

results in Figure 3c.

Re: We appreciate the insightful questions by the reviewer. The tumor targeting effect of RNP_{100%} can be attributed to the Enhanced Permeability and Retention (EPR) effect of tumor. Because of the leaky tumor blood vessels and lack of lymphatic drainage, the nanoparticles accumulated at tumor sites can be higher than that in the adjacent normal tissues. As seen in **Supplementary Figure 6a**, the tumor accumulation of RNP_{100%} increases along with the longer blood circulation time, leading to the enhanced imaging contrast with the surrounding normal tissues. However, because of the Always-ON design, the tumor-to-muscle ratio of RNP_{100%} was significantly lower than MRIN that can achieve pH-responsive Cy5 signal amplification (new **Supplementary Figure 6c**).

Line 154-157: It's worth noting that although both RNP_{100%} (Always-ON micelle) and MRIN exhibited good tumor targeting effect due to the enhanced permeability and retention (EPR) effect, MRIN achieved significantly higher tumor-to-normal tissue contrast due to the pH-responsive Cy5 signal amplification (Supplementary Fig. 6c).

Supplementary Figure 6. *In vivo* fluorescence images of mice after treatment with different micelles. (a) *In vivo* fluorescence images of the bilateral 4T1 tumor-bearing mice with or without 808 nm irradiation on right tumors at 3, 6, 12, and 24 h post-injection of different micelles. (b) The quantified extracellular percentages of different micelles *in vivo* at 3, 6, 12, 24 h post injection from the results of Supplementary Fig. 6a ($n = 4$). (c) The quantified Cy5 fluorescence ratio of tumor to adjacent normal tissues by the results of Supplementary Figure 6a ($n = 4$).

3. Does the principle total=intracellular + extracellular applicable to MRIN also apply to RNP_{0%}, RNP_{100%}? For RNP_{100%}, There is no acid response-mediated fluorescence recovery, so it is not certain that exhibited fluorescence means intracellular, extracellular nanoparticles also exhibited fluorescence. Similarly, the RNP_{0%} did not produce fluorescence, which does not mean that they were extracellular, the author needs to provide a reasonable explanation or correct the description in Figure 3b.

Re: Thanks for the reviewer's suggestion. We have added the related explanation in the revised manuscript. In our paper, we developed a pH/light dual-responsive monochromatic ratiometric imaging nanotechnology (MRIN) to dissect extracellular and intracellular distribution of nanoparticles in tumor tissues. In order to demonstrate the quantitative imaging feasibility of MRIN, RNP_{0%} (Cy5-labeled pH-insensitive micelle) and RNP_{100%} (Always-ON micelle) was established to simulate the artificial states of 0% and 100% endocytosis in vitro and in vivo. As the reviewer mentioned, it doesn't mean they are true 0% and 100% endocytosis. Because that the Cy5 signal of RNP_{0%} was completely 'OFF' whether cellular endocytosis or not due to its pH-insensitive nanostructure, so it could be used to simulate the artificial states of 0% endocytosis. And the Cy5 signal of RNP_{100%} was completely 'ON' due to the Always-ON design, so it could be used to simulate the artificial states of 100% endocytosis.

Line 114-116: These results demonstrated that RNP_{0%} and RNP_{100%} are suitable to simulate the artificial states of 0% and 100% endocytosis regardless of their real distribution in extracellular and intracellular compartments.

The caption of Figure 3b: The images of RNP_{0%} and RNP_{100%} before 808 nm irradiation were served as the extrapolated states of 0% and 100% internalization of nanoparticles.

4. Four kinds of particles, MRIN, RNP_{0%}, RNP_{100%}, and MRITN are used in the manuscript, and the self-assembly components include UPS, UPS-Cy5, UPS-Cy7.5, pH-insensitive polymer, pH-insensitive polymer-Cy5, pH-insensitive polymer-Cy7.5, UPS-Ce6. Although components and ratio were listed separately in the experimental steps, it is not very clearly stated in the text. The self-assembly composition and ratio of each particles may be listed in a table or in the corresponding figure.

Re: Thanks for the reviewer's good suggestion, we have listed the self-assembly composition and ratio of each particle in Supplementary table 1.

Supplementary Table 1. The polymer composition and their ratios for each nanoparticle.

Nanoparticle	Polymer 1	Polymer 2	Polymer 3	Ratio (%)
RNP _{0%}	PEH-Cy5 ₁	PEH-Cy7.5 ₃	Blank PEH	5:45:50
MRIN	UPS-Cy5 ₁	UPS-Cy7.5 ₃	Blank UPS	5:45:50
RNP _{100%}	UPS-Cy5 ₁	/	Blank UPS	5:95
MRITN	UPS-Ce6 ₁	UPS-Cy7.5 ₃	/	50:50

^a UPS is the abbreviation of ultra-pH-sensitive PEG_{5k}-*b*-P(DPA₄₀-*r*-EPA₄₀) copolymer.

^b PEH is the abbreviation of pH-insensitive PEG_{5k}-*b*-PEH₈₀ copolymer.

^c The right subscript of the dye represents dye conjugated numbers.

5. Why Cy7.5 also quenched Ce6? the author needs to provide FRET related explanation.

Re: Thanks for the reviewer's question. The overlap between the donor fluorescence emission spectra and the acceptor fluorescence excitation spectra is necessary for efficient FRET (Broussard JA, *et al. Nat Protoc* **2013**, 8, 265-81). In our study, Ce6 is the donor fluorophore and Cy7.5 is applied as acceptor fluorophore. As seen in **Supplementary Figure 11a**, there is good overlap between the emission spectrum of Ce6 and the excitation spectrum of Cy7.5, enabling the FRET effect from Ce6 to Cy7.5. In addition, we also have verified the FRET signal of MRITN (UPS-Ce6/UPS-Cy7.5 hybrid micelle) and UPS-Cy7.5 micelles at 630 nm for Ce6 excitation (**Supplementary Figure 11b**). The result showed that UPS-Cy7.5 micelle had no emission peak at 825 nm with excitation at 630 nm, while MRITN has two emission peaks at 670 nm and 825 nm, respectively, indicating that Cy7.5 can absorb the energy of Ce6 emission to emit its own signal. Therefore, the above results proved that FRET effect can occur between Ce6 and Cy7.5. We included the results in **Supplementary Figure 11**.

Line 218-219: Cy7.5 could quench the fluorescence and photosensitivity of Ce6 due to the FRET effect between them (Supplementary Fig. 11).

Supplementary Figure 11. The FRET effect between Ce6 and Cy7.5.

6. Regarding why PDT with intracellular and extracellular is better, the author needs to provide a mechanism explanation or further pathway data to support Figure 5.

Re: Thanks for the reviewer's suggestion. As seen in **new Supplementary Figure 14**, we found that MRITN bound to the cell membrane at physiological pH, and the cell binding ability of MRITN was significantly enhanced in a slightly acidic environment such as tumor microenvironment. Many studies have revealed that cell membrane-targeted PDT can lead to membrane dysfunction and disintegration only by a mild treatment (Liu L-H, *et al, Adv Funct Mater* **2017**, 27, 1700220; Kim J, *et al, J Control Release* **2014**, 191, 98-104). We also have proved that increased fluorescence of Ce6 and SOG was observed at cell membrane after photobleaching of Cy7.5 in **Figure 5a**. Therefore, the extracellular PDT relies on the damage to cell membrane for our *in vitro* cellular studies. While for the *in vivo* anti-tumor study, the extracellular PDT results from both the cell membrane-based PDT and destruction of extracellular matrix (ECM). What's more, the PDT dose (including photosensitizer dose and laser power) is one of the determinants for the anti-tumor efficacy of PDT. Currently, most activatable PDT rely on the intracellular exposure to exert lethal tumor damage, while a majority of nanophotosensitizers distributed in the extracellular space of the tumor site are useless (Zhou S, *et al, Angew Chem Int Ed Engl* **2020**, 59, 23198-23205; Lovell JF, *et al, Chem Rev* **2010**, 110, 2839-2857). Based on MRITN technology, we can light up the extracellular nanophotosensitizer to specifically increase the efficient photosensitizer dose in tumor, leading to significantly better tumor inhibition than intracellular PDT.

Line 228-231: Damage to the cell membranes plays an important contribution to the extracellular PDT. We found that MRITN could bind to the cell membrane at physiological pH, and the cell binding ability of MRITN was significantly enhanced in a slightly acidic environment such as tumour microenvironment (Supplementary Fig. 14)

Supplementary Figure 14. The cell membrane binding ability of MRITN at different pH values. The MRITN was activated by 808 nm irradiation before cell treatment.

REVIEWERS' COMMENTS

Reviewer #2 (Remarks to the Author):

I suggest to accepting the revised manuscript.

Point-by-point response to reviewers

We would like to thank the reviewers for the insightful and constructive comments! Below is a list of the point-by-point responses to the reviewer comments.

Reviewer #2 (Remarks to the Author):

I suggest to accepting the revised manuscript.

Re: We thank the reviewer for the very supportive comments.